# A Lightweight Fault-Detection Scheme for Resource-Constrained Solar Insecticidal Lamp IoTs

**DOI:** 10.3390/s23156672

**Published:** 2023-07-25

**Authors:** Xing Yang, Lei Shu, Kailiang Li, Edmond Nurellari, Zhiqiang Huo, Yu Zhang

**Affiliations:** 1College of Engineering, Nanjing Agricultural University, Nanjing 210031, China; harryyangx@gmail.com; 2College of Artificial Intelligence, Nanjing Agricultural University, Nanjing 210031, China; kailiang_li@njau.edu.cn; 3College of Engineering, University of Lincoln, Lincoln LN6 7TS, UK; enurellari@lincoln.ac.uk; 4Department of Population Health Science, King’s College London, London SE1 8WA, UK; zhiqiang.huo@kcl.ac.uk; 5Department of Aeronautical and Automotive Engineering, Loughborough University, Loughborough LE11 3TU, UK; y.zhang@lboro.ac.uk

**Keywords:** distributed fault detection, solar insecticidal lamps internet of things, quantile method, two-hop information

## Abstract

The Solar Insecticidal Lamp Internet of Things (SIL-IoTs) is an emerging paradigm that extends Internet of Things (IoT) technology to agricultural-enabled electronic devices. Ensuring the dependability and safety of SIL-IoTs is crucial for pest monitoring, prediction, and prevention. However, SIL-IoTs can experience system performance degradation due to failures, which can be attributed to complex environmental changes and device deterioration in agricultural settings. This study proposes a sensor-level lightweight fault-detection scheme that takes into account realistic constraints such as computational resources and energy. By analyzing fault characteristics, we designed a distributed fault-detection method based on operation condition differences, interval number residuals, and feature residuals. Several experiments were conducted to validate the effectiveness of the proposed method. The results demonstrated that our method achieves an average F1-score of 95.59%. Furthermore, the proposed method only consumes an additional 0.27% of the total power, and utilizes 0.9% RAM and 3.1% Flash on the Arduino of the SIL-IoTs node. These findings indicated that the proposed method is lightweight and energy-efficient.

## 1. Introduction

The solar insecticidal lamp (SIL) has gained widespread adoption in agricultural pest management and control, offering an environmentally friendly approach to pest control. Recent advancements in IoT technology have enabled SILs to expand their functionalities and improve operational life through pest monitoring, pest outbreak area positioning, and energy optimization in battery-powered devices [1]. Yang et al. [2] have indicated that the fixed effective killing distance of SIL ranges from 50 to 110 m, which falls within the communication range of ZigBee. Leveraging this characteristic, SIL-IoTs nodes can collect and transmit data related to pest statistics (e.g., the number of pests killed in a short period of time), component status information (e.g., voltage and current values of various components), and meteorological environment information to the back-end system via the network [3]. This data transmission allows farmers to accurately use pesticides in areas with varying pest populations, therefore avoiding excessive pesticide usage, as shown in Table 1. Moreover, IoT devices facilitate continuous and remote monitoring of SIL-IoTs’ component status, enabling timely failure reporting and improving the reliability and data quality of SIL-IoTs.

Figure 1 illustrates some key elements and functionalities of a typical SIL-IoTs node. Among other core components, sensors are used to further embed various intelligence capabilities into the SIL-IoTs node. For example, a solar energy system allows the SIL-IoTs node to be charged during the day, while at night it is programmed to automatically attract pests. A metal mesh is used to kill pests (by contact) by discharging a sudden high-voltage pulse. During this process, several intelligent sensors monitor environmental conditions, calculate the number of pests killed and determine the operating status of the modules. During rainy periods, the SIL-IoTs switch to sleep mode by turning off the lure lamp and metal mesh to prevent damage and save energy.

Typically, SIL-IoTs nodes are geographically dispersed and deployed in an unattended and harsh environment. Inevitably, the SIL-IoTs nodes are susceptible to aging, theft, and vandalism [5]. According to several relevant literature [6], there have been 19 related news reports of SIL failures in the past 20 years, and a total of more than 7000 SILs have been abandoned due to insufficient fault detection and maintenance work, which is not conducive to the promotion of products and the establishment of user confidence.

The above issues result in faulty conditions and abnormal operation of SIL-IoTs nodes, which affect the operational capabilities and overall performance of SIL-IoTs. For instance, if the energy harvesting system fails (causing the solar panel to continuously charge the battery without a control mechanism), the battery will eventually heat up and cause performance degradation, or even explode and cause damage to SIL-IoTs nodes. In addition, the deployment of SIL-IoTs nodes in remote locations makes real-time inspection and maintenance difficult. Therefore, it is a challenging task (to monitor and detect the SIL-IoTs node faults) to ensure adequate and efficient operation throughout the lifecycle. If there is an adequate provision of computational capacity and energy, traditional approaches can provide good detection performance in terms of real-time response, data loss prevention, and less data transmission [7,8].

The motivation and benefits of this research are as follows:

As SIL-IoTs nodes are often deployed in the field, they are susceptible to aging, vandalism, and other factors that can lead to failures. To detect faults in SIL-IoTs, appropriate fault diagnosis methods need to be investigated. Deploying fault diagnosis methods on the device side can improve the efficiency of device data usage and reduce the energy consumption of missing data and transmitted data due to data backhaul. The background characteristics of SIL-IoTs need to be considered when designing fault diagnosis methods, including:The computational burden of fault-detection strategies needs careful consideration in practical applications. For example, SIL-IoTs nodes are resource-constrained devices, which indicates that the fault-detection model should be lightweight to reduce the computational burden;The low deployment density of SIL-IoTs node leads to an insufficient number of nodes in geographical proximity, and the existing distributed fault diagnosis methods are difficult to achieve better results in this case, hence it is critical to design a distributed fault diagnosis method with low dependence on the number of neighboring nodes.

SIL-IoTs is a kind of typical agricultural IoT equipment, thus the proposed method in this paper can also be used in IoT equipment with similar characteristics in, e.g., intelligent irrigation equipment, and micro weather stations.

Based on the above, this research makes the following contributions:We propose a novel and easily implementable fault-detection scheme for SIL-IoTs nodes deployed in low-density fields. This scheme is based on multi-factor correlation analysis, ensuring high performance even in scenarios where relevant data from neighboring nodes are missing or only a small number of neighboring nodes are operational;We develop a computationally efficient method for estimating weight parameters in linear regression using historical data to mitigate the limited computational capability and bandwidth. This approach reduces the computational burden while maintaining accurate fault-detection capabilities;We introduce a regression-based machine health prediction method to deal with the impact of unreliable neighboring nodes on fault-detection probability. This approach leverages and combines results from multiple neighboring nodes, enhancing the reliability and robustness of fault detection.

These contributions address key challenges in fault detection for SIL-IoTs, such as handling missing data, optimizing computational resources, and improving reliability in the presence of unreliable neighboring nodes. Thus, this research contributes to the advancement of fault-detection techniques for SIL-IoTs in agricultural settings.

## 2. Related Work

Fault detection and prediction are critical to enabling proactive intelligent device health management [9,10]. A well-established approach is to detect faults in a centralized manner at the server level, which requires periodic collection of information from all nodes (i.e., each SIL-IoTs periodically transmits to the data collection server) and performing inference processes at the back end [7]. For instance, the connectivity metrics of all the nodes are transmitted to the back end and the root causes are troubleshot using a decision tree [11]. Tang et al. [12] proposed a neighborhood hidden conditional random field method to monitor the health of wireless sensor networks. The posterior probability of different faulty states is estimated and used to classify faults at the back end.

As shown in Table 2, unlike established and traditional IoT applications, SIL-IoTs devices are mainly characterized by (1) limited on-board storage and computing capacity, (2) remote deployment locations with poor network conditions, and (3) deployment to cover a large geographical area. Due to the high communication overhead and detection delay caused by multi-hop data transmission, this approach is not efficient in terms of both overall detection performance and resource allocation (i.e., devices are battery-powered and therefore have limited energy). Although Yang et al. [4] has proposed a scheme for fault self-inspection in the Arduino chip of SIL-IoTs, the scheme does not take into account the information interaction between nodes, and further analysis cannot be performed for some fault situations, such as the mismatch between the current and light intensity of the solar panel.

Since SIL-IoTs operate in multiple interrelated ways, the distributed fault-detection strategy, which detects faults via local evidence on sensor nodes, can be applied to address these issues [5]. Furthermore, the distributed fault-detection methods in wireless sensor networks (WSNs) need to consider the computational capacity, bandwidth usage, and residual energy of nodes [22]. Therefore, the relevant literature work on such distributed fault-detection methods is worthy of reference.

Several contributions have been made over the last two decades. One of the earliest attempts can be found in [13], where consistency between local components is modeled to detect faults in discrete-event systems. In contrast to [13], Chen et al. [14] proposed a distributed fault-detection (DFD) method for measurements of WSNs by checking the number of faulty states of neighboring nodes calculated by residual analysis between neighboring nodes. In [15], a similar but slightly improved method is proposed where each node detects faults by checking the number of neighboring nodes in possibly normal states, which can be obtained by the method proposed in [14]. The results in [15] indicate that the improved method can be applied in WSNs with fewer neighboring nodes.

In [14,15], the detection threshold is predefined according to different applications at the time of deployment, which is a design parameter and highly dependent on the application and requires specific knowledge. To avoid the need for on-site technical expertise, Panda and Khilar [16] proposed a distributed self-fault-detection (DSFD) method for large-scale WSNs, where each WSNs node can identify its own faulty conditions via a modified three-sigma edit test.

The sliding window is an alternative method for detecting faults. For example, the TinyD2 method [7] has been proposed to detect faults by first calculating a cumulative sum on a sliding window. The original values are then reordered using the bootstrap method to generate a new data sequence. If a change is detected, the faulty node is identified. In addition, the TrusDet method [19] detects faults using a fused result from a sliding window, where a more recent data point has a greater influence on the data fusion. A vote is then taken to determine the status of the current area. All these approaches can be performed on sensor nodes and require few parameters. However, fault detection based on node voting results will fail if more than half of the nodes fail. In addition, their performance is affected by the number of neighboring nodes and will fail if neighboring nodes are not correlated with the target node.

Recent research has focused on correlation analysis-based fault-detection schemes, which are suitable for optimal fault detection and are characterized by their independence from expert knowledge. For instance, Hou et al. [23] applied the Jennic JN5139 sensor board and controller board to fuse decisions evaluated by three sensor nodes in a motor monitoring system. In [17,24], the spatial correlation analysis-based fault-detection methods are developed to compress the data transmitted by neighboring nodes that affect the target node. Fu et al. [20] proposed a trend correlation-based fault detection (TCFD) method, which detects faults via trend correlation analysis and the mean value of neighboring nodes. The self-starting mechanism is designed to reduce the response time of nodes to faults. In addition, Cheng et al. [25] applied space–time correlation analysis to estimate the weight value for fault detection, resulting in high detection accuracy and low false alarm rate for temperature, humidity, and voltage data. Unlike [17,24,25], Liu et al. [26] proposed a metric correlation-based distributed fault-detection method (MCDFD), which is motivated by the fact that abnormal correlations between measurement metrics indicate faults. By analyzing the metric correlation between sensor readings, the MCDFD method can reduce communication overhead and has high detection accuracy under conditions of dense distribution and high node failure rate.

In summary, the advantages of recent studies include (1) avoiding large amounts of data transmission to the back end using local information decision-making, and (2) avoiding inaccurate fault diagnosis results due to missing or asynchronous data from neighboring nodes when fault diagnosis is performed in the back end. It should be noted that recent studies in Table 2 are based on scenarios with a high deployment density of sensor nodes, whereas the deployment density of SIL-IoTs nodes is usually sparse [2], which denotes that diagnosing fault by voting strategy can lead to a decrease in diagnostic accuracy. The literature [2] shows that when the effective pest-killing range of SIL-IoTs nodes is 110 m (i.e., deploying SIL-IoTs nodes at 110 m intervals), only 10 nodes need to be deployed on a 600 m × 600 m map according to the optimal deployment method proposed in the literature. Compared to the literature [18], which deploys 1024 nodes on a 512 m × 512 m map, or the literature [16], which deploys 1024 nodes on a 1000 m × 1000 m map, the deployment density of SIL-IoTs nodes is significantly lower. In addition, distributed fault diagnosis methods require data interaction between nodes, which generates additional communication energy consumption, which is detrimental for SIL-IoTs nodes.

Based on the above, the reviewed methods can be analyzed based on their adaptability to detection thresholds and their method complexity. The adaptability of the detection threshold allows nodes to set appropriate fault-detection thresholds based on the environment and component status, thus improving the accuracy of the fault-detection algorithm across different nodes. The proposed method in this study detects faults by comparing two fault-related features instead of relying on predefined thresholds. On the other hand, the complexity of a method serves as an indicator of its practicality in nodes with limited resources. The proposed method presented in this research demonstrates low complexity by storing only a few parameters in the Arduino of SIL-IoTs nodes and utilizing summation calculations for fault detection. This ensures that the fault-detection method remains practical and feasible even in resource-constrained nodes. By considering adaptability to detection thresholds and method complexity, the proposed method offers a promising solution for fault detection in SIL-IoTs, providing improved accuracy and practicality in agricultural settings with limited resources.

## 3. System Model

### 3.1. SIL-IoTs System

The SIL-IoTs system consists of *N* nodes operating in a cooperative environment to transmit data and make local decisions, where N=1,2,…,N. The *i*th node is equipped with both control and sensor data processing capabilities (see Figure 2).

Control signals are used to switch the SIL-IoTs on and off in a scheduled and optimal manner. For instance, to protect the battery, the solar charge controller will stop charging the solar panel when the battery is fully charged. It also cuts off the power supply when the remaining energy in the battery is low. The lure lamp and metal mesh are only switched on during the night and when it is not raining (the switch-on time is a designed parameter, but will generally be between 7 p.m. and 4 a.m.). In addition, the lure lamp and metal mesh components are switched off if a fault is detected or if they are damaged. This simple on/off approach can be easily implemented on a low-power microcontroller (e.g., Arduino with 20 MHz CPU speed, 32 KB program memory size, and 2 KB RAM size).

Data collected from several on-board sensors are used to monitor the state of SIL-IoTs nodes and obtain statistical indicators that contribute to the estimation of pest occurrence, energy consumption trends, and fault symptoms [5]. Specifically, the voltage and current values of the battery, solar panel input/output, lure lamp, and metal mesh (represented as $V^{B}$, $C^{B}$, $V^{S}$, $C^{S}$, $V^{L}$, $C^{L}$, $V^{M}$, and $C^{M}$) are key contributors to SIL-IoTs energy management and module monitoring. Voltage pulse count (the number of high-voltage pulses released by the metal mesh, represented as $V^{C}$) and sound count (a sharp noise when pests contact the metal mesh, represented as $S^{C}$) are used to estimate the number of pests killed, which helps to establish pest occurrence statistics. In addition, meteorological observations (i.e., light intensity, air temperature, and related humidity, denoted as *L*, $T_{out}$, and *H*) are used to monitor the environmental conditions of SIL-IoTs. Finally, the temperature difference of the SIL-IoTs device, which is obtained by the temperature difference between the temperature inside the enclosure (denoted as $T_{in}$, box temperature sensor in Figure 1) and the temperature outside the enclosure (denoted as $T_{out}$, air temperature sensor in Figure 1), is used to estimate the thermal state of the battery and IoTs modules inside the electrical circuit (enclosure).

### 3.2. Fault Types

The main purpose of this paper is to detect the fault, which cannot be found without the information interaction of SIL-IoTs nodes and their neighboring nodes. Since only one piece of node information is considered, the root cause that leads to the mismatch of two measurements cannot be found in our previous research [4]. Based on this, we aim to detect the following faults:

The mismatch between *L* and $C^{S}$ (known as F1): can be expressed in (Equation 1) according to [4]. There may be a fault in the light intensity sensor or the solar panel which can be detected by the neighboring information. The fault of the light intensity sensor may lead to an error in the estimation and prediction of energy harvesting. The power generated by the solar panel at the corresponding light intensity value is usually used to evaluate the energy conversion of the solar panel [27,28]. Therefore, the fault of the solar panel may fail the monitoring of the module.
(1)$$F1_{state}=\left\{\begin{array}{cc} |0.0316\times L-6.28-C^{S}|⩾450, & L<10^{6}\\ |3200-C^{S}|⩾450, & L⩾10^{6} \end{array}\right.$$

The mismatch between $T^{0}$ and $T^{1}$ (known as F2): is represented in (Equation 2) according to [4]. The battery and the IoTs device in the electrical box may be in a thermal state due to some faults, causing a large temperature difference between the two. In this case, it is important to assess as soon as possible whether the problem is caused by a sensor fault or by heat generation. If a sensor fault is the cause, a recalibration or reboot is required. Otherwise, power to the node should be removed and maintenance personnel should be notified.
(2)$$F2_{state}=\left\{\begin{array}{cc} \mathrm{normal}, & |T^{1}-T^{0}|<9\\ \mathrm{abnormal}, & |T^{1}-T^{0}|⩾9 \end{array}\right.$$

SIL not switched on according to schedule (known as F3): On the one hand, when a clock chip fault occurs, the local time is not synchronized with the background time (e.g., the clock chip is restarted due to lack of power resulting in an abnormal local time), which will cause the SIL to turn on at a non-setting time. In this case, the SIL is likely to be switched on during the day and off at night. On the other hand, the nighttime is also estimated by the light intensity sensor reading being close to 0 at night and the SIL being on (both with current values significantly greater than 0). When the light intensity sensor value is significantly greater than 0 and the lure lamp and high-voltage metal mesh are on, it is not possible to determine whether the light intensity sensor is faulty causing an abnormal reading or the clock chip is faulty causing the local time to be abnormal. Such faults can be identified by the light intensity values of neighboring nodes and the on/off status of the SIL. When the light intensity values of neighboring nodes are close to 0 and the target node has a high light intensity value, the light intensity sensor reading is considered to be abnormal. In the absence of rain and with sufficient power remaining, the clock chip of the target node can be identified as having an abnormal reading when the neighboring nodes are off and the target node is on.

## 4. Proposed Method

The proposed method is triggered when the *i*th node cannot detect a fault using its own available local information. In addition, we assume that the measurements of the target node (node under fault detection) and its neighboring node (nodes geographically adjacent to the target node) are time synchronized.

The flowchart of the proposed scheme is illustrated in Figure 3. To detect the above faults, this section proposes distributed fault-detection methods for operating condition differences, interval numbering residuals, and feature residuals.

Since the proposed distributed fault-detection method is implemented in the Arduino and only needs to store the bounds of the interval between the light intensity and solar panel current values and the thresholds for judging the on/off of the lure lamp, the proposed method requires fewer parameters for its computation. In addition, the proposed method only involves simplified logic operations and judgments, so the computational complexity and running time are relatively low, which contributes to the reduction of energy consumption. Based on the above, the proposed distributed fault-detection method has the advantages of light weight, energy saving, and reduced dependence on fault-detection thresholds. Therefore, the method is executed relatively infrequently and does not result in any ongoing additional communication overhead and communication energy consumption due to the use of two-hop information for distributed fault detection.

### 4.1. Correlation Analysis

When performing distributed fault detection, it is first necessary to determine whether the features between neighboring nodes are spatially correlated (i.e., whether the nodes have the same feature trend over time), and distributed fault detection can only be performed if the features have a high spatial correlation. This section uses a widely used spatial correlation analysis method, the Pearson correlation coefficient (PCC) [29]. The Pearson correlation coefficient is an accepted and valid indicator of correlation analysis (expressed as *r*), ranging from −1 (highly negative correlation) to +1 (highly positive correlation) [30]. For two variables *X* and *Y* of a given sample size *n*, *r* can be expressed as:(3)$$r=\frac{\sum_{k=1}^{n}\left(X_{k}-\bar{X}\right)\left(Y_{k}-\bar{Y}\right)}{\sqrt{\sum_{k=1}^{n}{\left(X_{k}-\bar{X}\right)}^{2}}\sqrt{\sum_{k=1}^{n}{\left(Y_{k}-\bar{Y}\right)}^{2}}}$$
where $\bar{X}$ and $\bar{Y}$ denote the average of *X* and *Y*. The spatial correlation between the device to be detected and its neighboring nodes can be obtained from Equation (Equation 3), which represents the covariance of *X* and *Y* divided by the product of the standard deviations of *X* and *Y*. Fault detection can be performed based on the cumulative sum of the residuals of the correlation features of the neighboring node and the faulty node only if there is a significant positive spatial correlation between the faulty node and its two-hop neighboring nodes.

In addition, since the proposed method detects mismatch faults by comparing the cumulative sum of residuals between multiple features related to the fault, it is necessary to determine whether there is a feature correlation between multiple features. If the degree of correlation between the two features is low, then detection using the cumulative sum of residuals is less reliable. The Pearson correlation analysis is difficult to ensure good results because the units of the feature metric may be different. In this case, a correlation analysis method that can ignore the units of the features is required.

Random forest methods have significant advantages in analyzing feature importance using small samples [31]. To select important features, a Permutation Importance score (PI) is calculated for each decision tree of the random forest [32]. PI is obtained by randomly shuffling each feature and computing the change in the performance of the random forest. As shown in Equation (Equation 4), the importance score ranking is estimated by differences between the regression accuracy without randomly exchanging permuted out-of-bag data (denoted as $E_{k}$) and the regression accuracy with randomly exchanging permuted out-of-bag data (denoted as $E_{xk}$). *n* denotes the number of decision trees included in the random forest. Based on this, features are reordered from largest to smallest in the ranking importance score.
(4)$$PI=\frac{1}{n\sum_{k=1}^{n}\left(E_{xk}-E_{k}\right)}$$

### 4.2. Operating Condition Difference Based Fault Detection

For features that vary significantly between operating conditions, labels can simply be set based on the operating condition. As shown in Figure 4, the current values of the lure lamp and the high-voltage metal mesh increase significantly when the SIL is in operation, therefore, a threshold can be set to set different labels for the on and off conditions based on historical data. For the time state (which only distinguishes between day and night), the light intensity sensor values can be used to determine this. If the light intensity value is less than 10 Lux, the current time is judged to be night, and vice versa for day. Based on this, when detecting a clock chip fault, the faulty node can be judged according to the operating status label of the neighboring node’s switch lights, without having to calculate the difference between the current value of the faulty node and the neighboring node, thus reducing the amount of data transmission and the amount of fault-detection calculations.

There are two possibilities for a clock chip fault, one scenario is an abnormal light intensity value and the other is an abnormal clock chip data. A clock chip fault will result in abnormal local data which will affect the on/off status of the time-controlled SIL. Specifically, the clock chip fault is detected based on the clock chip abnormality when the node light intensity value is greater than 10 Lux and the lure lamp or high-voltage metal mesh is turned on and estimated by the current value.

Since the operating condition of SILs can be simply classified as either on or off, day and night can be indicated by whether the light intensity value is significantly greater than 0. Therefore, the operating condition of the clock chip fault can be indicated by the current value of the lure lamp (denoted as $C^{M}$), the current value of the high-voltage metal mesh (denoted as $C^{L}$), and the light intensity value L, as shown in Algorithm 1. $T^{W}$ represents the operating condition number of the SILs. The lure lamp and high-voltage metal mesh will only be switched on if the node determines that it is currently nighttime based on the local time data. At this time, $C^{M}$ should be greater than 60 mA and $C^{L}$ should be greater than 600 mA. Therefore, $T^{W}$ will be set to 1 and 0 otherwise. $T^{L}$ represents the day and night conditions characterized by the light intensity value, stored as 0 and 1. 1 means that it is currently day based on the light intensity value and 0 means that it is currently night based on the light intensity value. As SIL-IoTs nodes are deployed in agricultural fields, they should not be able to receive illumination from external light sources such as streetlamps at night, and their light intensity value should be at a lower level. Therefore, when the light intensity value is below 10 Lux, it is judged to be currently in a night state.
**Algorithm** **1:** Calculating the operating condition $T^{W}$ and the time condition $T^{L}$.
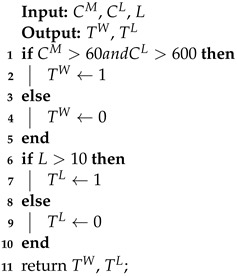


The faulty node performs distributed fault-detection through the process of Algorithm 2 after obtaining the SIL operating condition and the day or night number based on the light intensity value of its fault-free two-hop neighboring node. $T_{y}^{W}$ and $T_{y}^{L}$ represent $T^{W}$ and $T^{L}$ of the node *y* (faulty node). $S\left(T^{W}\right)=\left\{T_{1}^{W},\ldots ,T_{K}^{W}\right\}$, $S\left(T^{L}\right)=\left\{T_{1}^{L},\ldots ,T_{K}^{L}\right\}$ denote the $T^{W}$ and $T^{L}$ of all two-hop neighboring nodes of the node *y*, where *K* indicates the number of two-hop neighboring nodes of the node *y*. $S_{T^{W}}$ and $S_{T^{L}}$ denote the accumulated sum of $T^{W}$ and $T^{L}$ residuals of the faulty node and its two-hop neighboring nodes, where the initial values of both $S_{T^{W}}$ and $S_{T^{L}}$ are set to 0. The final value of $S_{T^{W}}$ is obtained by accumulating the absolute value of the residual value between $T_{y}^{W}$ and $T_{i}^{W}$ (each two-hop neighboring node of node *y*). The final value of $S_{T^{L}}$ is obtained in the same way. When $S_{T^{W}}$ exceeds $S_{T^{L}}$, the abnormal clock chip data fault is detected and F3−W is uploaded to the back end. When $S_{T^{W}}$ is less than $S_{T^{L}}$, the light intensity value is abnormal, and the fault code F3−L is uploaded to the back end. When $S_{T^{W}}$ is equal to $S_{T^{L}}$, the result is uncertain, and the corresponding fault label is F3−U.
**Algorithm** **2:** Fault detection according to $T^{W}$ and $T^{L}$.
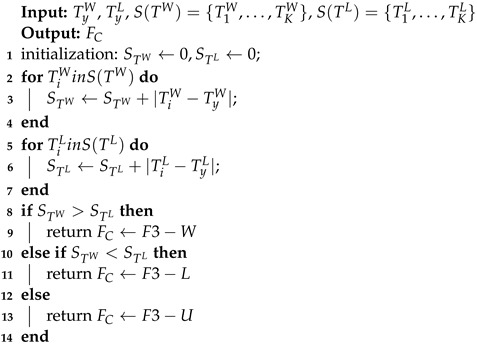


### 4.3. Interval Numbering Residuals Based Fault Detection

Not all faults have significant differences in operating conditions for the relevant features. When there is no significant difference in operating conditions, it is necessary to compare the data differences between the two features. Due to adverse factors, e.g., environmental differences, deviations in sensor readings, aging of devices and welding processes, there may be large differences in the same features at the same time by neighboring nodes, making it difficult to detect fault by residuals between the faulty and neighboring node features directly. For instance, the light intensity value and solar panel current value are affected by the degree of the dust cover, sunlight irradiation angle, device installation location, and the degree of aging of the device. The light intensity value or solar panel current value of different nodes under the same climatic environment and lighting conditions may have large differences. As shown in Figure 5, although there are differences in the relevant feature values of different nodes, the change trends of the feature values at the same time are similar. Therefore, the historical data can be used for dimensionless processing, i.e., the current feature value is estimated to be in what interval in the historical data. Based on this, the proposed method sorts the features that fit this scenario based on the historical data and sets the segment intervals, so that when a new feature value is obtained, it is estimated to fall within a certain interval, completing the dimensionless processing of the feature value. In this way, the type of fault can be determined by estimating the residuals of the interval numbers of the corresponding features of the faulty node and its neighboring nodes.

To quantify the distribution patterns of fault-related features at different nodes and to reduce the storage of relevant parameters, the proposed method uses the quantile method to construct mapping intervals. This method divides the range of probability distribution of a random variable into multiple equal parts of numerical points and is commonly used as median, quartile, percentile, etc. [33]. To take the quartile method as an example, suppose a set of data $X=x_{1},x_{2},x_{3},\ldots ,x_{n}$, where *n* denotes the number of data. After sorting them in ascending order, choose $Q_{1}=1+\left(n-1\right)*0.25$, $Q_{2}=1+\left(n-1\right)*0.5$, and $Q_{3}=1+\left(n-1\right)*0.75$ as quartiles to divide *X* into four segments as shown in Figure 6. $A_{1}$, $A_{2}$, $A_{3}$ and $A_{4}$ represent the quartiles based on the minimum (Min), $Q_{1}$, $Q_{2}$, $Q_{3}$ and maximum (Max) constituting the four equal intervals, based on which each piece of data in X can be mapped to the corresponding interval as shown in Equation (Equation 5).
(5)$$f=\left\{\begin{array}{cc} A_{1}, & when\;Min\leq x_{i}\leq Q_{1}\\ A_{2}, & when\;Q_{1}\leq x_{i}\leq Q_{2}\\ A_{3}, & when\;Q_{2}\leq x_{i}\leq Q_{3}\\ A_{4}, & when\;Q_{3}\leq x_{i}\leq MAX \end{array}\right.$$

The quantile method is widely used in engineering applications due to its simplicity and ease of use. However, current quantile-based distributed fault-detection methods do not set intervals based on historical data but rather perform fault detection based on multiple data from neighboring nodes simultaneously. For example, the process of the quantile fault-detection method proposed in the literature [34] consists of:

Step 1: Collect information about neighboring nodes. Suppose the neighboring nodes of node $S_{i}$ are $N\left(S_{i}\right)=S_{i}1,S_{i}2,\ldots ,S_{i}k$, then the dataset of neighboring nodes of node $S_{i}$ is $X\left(S_{i}\right)=x_{i}1,x_{i}2,\ldots x_{i}k$.

Step 2: Sort $X(S_{i})$ in descending order, extract the $Q_{1}$, $Q_{2}$, and $Q_{3}$ values, and calculate the difference between the value of each neighboring node and the median based on this, as shown in Equation (Equation 6).
(6)$$d_{i}=x_{i}-Q_{2}$$

Step 3: Normalize data according to the $Q_{1}$, $Q_{2}$ and $Q_{3}$ values and $d_{i}$, as shown in Equation (Equation 7).
(7)$$y_{i}=\frac{d_{i}}{Q_{3}-Q_{1}}$$

Step 4: Compare the normalized value with the set threshold value θ and a fault condition is judged when the threshold range is exceeded, as shown in Equation (Equation 8).
(8)$$f_{S_{i}}=\left\{\begin{array}{cc} Fault\;free, & when\;|y_{i}|\;\leq \theta \\ Fault, & when\;|y_{i}|\;>\theta \end{array}\right.$$

If the number of neighboring nodes is less than four, the quadrature fault-detection method will no longer work. Considering the low deployment density of SIL-IoTs nodes, where each node may have only one or two two-hop neighboring nodes. Therefore, the quartile method is unlikely to be effective. Since the data collected by SIL-IoTs nodes typically follows a historical cycle pattern, the proposed method uses the quantile method for segmentation of historical data, mapping the currently collected data to the corresponding interval numbers, as shown in Equation (Equation 5). Based on this, a mismatch fault between the solar panel current value (denoted as $C^{S}$ below) and the light intensity value (denoted as *L* below) can be detected by the following process, assuming that the faulty node $S_{i}$ has at least one neighboring node that does not have an associated fault:

Step 1: The quantile method is used to obtain the quantile points of the fault-free history data, and to generate segment intervals and numbers, as shown in Algorithm 3, where $A(C^{S})$, A(L), $C(C^{S})$, and C(L) indicate segment intervals of $C^{S}$ and *L* and numbering of $C^{S}$ and *L*, respectively. The “sort()” function is to sort the set in ascending order, $Q_{i}$ indicates the i−th quantile, and the “Min()” and “Max()” functions are used to take the minimum and maximum values of the set, respectively, “C()” means the interval number, *n* means the amount of historical data, and *m* is the number of segment intervals.
**Algorithm** **3:** Conventional method of establishing $C^{S}$ and *L* segment intervals and numbering for each node from historical data.
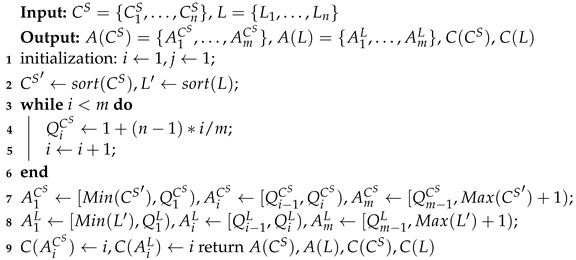


   However, Arduino cannot store long-term historical data, thus it is necessary to iterate through earlier data at the node side to calculate the quantile points, and the method process is shown in Algorithm 4. Taking the light intensity value as an example, the initialized interval set $A_{ini}\left(L\right)$$=A_{\left(ini,1\right)}^{L},\ldots ,A_{\left(ini,m\right)}^{L}$ is first set for all nodes based on the experience of the earlier historical data. The set $L_{ini}=C_{1},\ldots ,C_{m}$, a statistical set of initialized number of intervals, consists of *m* zeros and is used to count the number of corresponding intervals to which all sampled values belong during the iteration, where *m* denotes the number of intervals. The solar panel current value is a fixed value of 3200 mA when the light intensity is greater than 100,000 Lux, thus no count is performed when the light intensity is greater than 100,000 Lux. The number of times is therefore not counted for light intensity values above 100,000 Lux. When the sensor collects the latest light intensity value $L_{t}$, it determines that $L_{t}$ belongs to the interval range $A_{\left(ini,i\right)}^{L}$ of $A_{ini}\left(L\right)$, and adds one to the corresponding $C_{i}$ value in $L_{ini}$, thus counting the data distribution of light intensity values during the iteration.

When the light intensity value is below 100 Lux several times (i.e., when the sun sets), the sum of all interval statistics $L_{s}$ and the interval of quantile index value $L_{a}$ are counted for that day, to calculate each quantile $Q^{L}$. As shown in Figure 7, when calculating the quantile $Q_{2}^{L}$, the lower limit of interval statistics $A_{l}$ and the upper limit $A_{r}$ are initialized to 0. When $A_{r}$ is smaller than $2L_{a}$, the value of $A_{r}$ is assigned to $A_{l}$ and the value of $A_{r}$ becomes $A_{r}^{\prime }=A_{r}+C_{1}$. When $A_{r}^{\prime }$ is greater than $2L_{a}$, the next step is judged; otherwise, the $A_{r}^{\prime }$ value is assigned to $A_{l}$ and the $A_{r}^{\prime }$ value becomes $A_{r}^{\prime \prime }=A_{r}^{\prime }+C_{2}$ until the latest $A_{r}$ value is greater than $2L_{a}$. Based on this, the $Q_{\left(ini,2\right)}^{L}$ index is determined to be closer to $A_{l}$ or $A_{r}$ to calculate $Q_{2}^{L}$ more accurately. $A_{r}$ in Figure 8 is closer to $2L_{a}$, so the upper bound of $A_{\left(ini,2\right)}^{L}$$A_{\left(ini,2\right)}^{L}\left(r\right)$ is used to calculate $Q_{2}^{L}$. Since each interval contains $L_{a}$ data equally, the calculation can be based on the interval group distance, i.e., $Q_{2}^{L}=A_{\left(ini,2\right)}^{L}\left(r\right)\frac{2L_{a}}{A_{r}}$.
**Algorithm** **4:** The *L* segment interval and numbering of each node is established by iterations of the earlier data, and $C^{S}$ is the same.
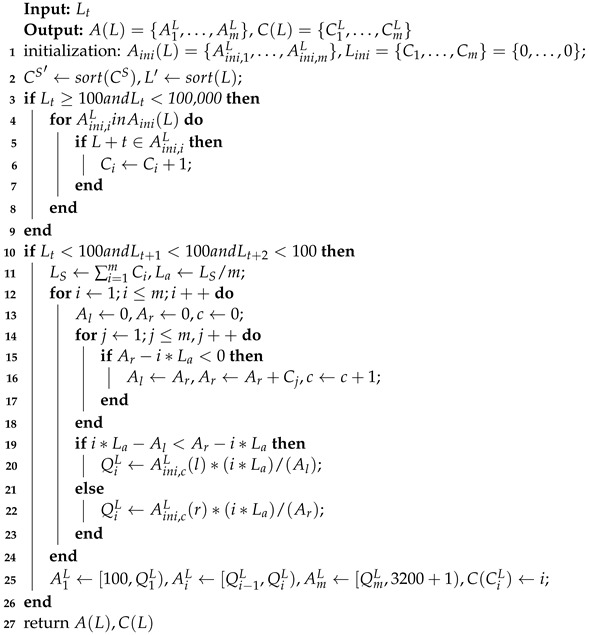


Step 2: When there is a mismatch between the light intensity value and the solar panel current value at the faulty node $S_{i}$, node $S_{i}$ first calculates its own $C^{S}$ and *L* corresponding interval numbers $N_{S_{i}}^{C^{S}}$ and $N_{S_{i}}^{L}$ according to Algorithm 5. Second, the two-hop neighboring node is identified by a secondary broadcast, and the two-hop neighboring nodes calculate their own $C^{S}$ and *L* corresponding interval numbers $N_{n}^{C^{S}}$ and $N_{n}^{L}$, respectively, and send them to node $S_{i}$. It is notable that the two-hop neighboring nodes first need to make sure that their solar panels and light intensity sensors are not open-circuit; otherwise, it does not participate in the distributed fault detection.

Step 3: As shown in Algorithm 6, the F1 fault is detected by the accumulated residuals of the collected interval numbering values of the two-hop neighboring nodes and the interval numbering values of the faulty nodes. *k* denotes the number of two-hop neighboring nodes without associated faults, $\theta_{C^{S}}$ denotes a preset $C^{S}$ interval numbering deviation threshold, and $\theta_{L}$ denotes a preset *L* interval numbering deviation threshold. $\theta_{C^{S}}$ and $\theta_{L}$ is used to determine whether there is a significant difference between the interval numbering of the faulty node and the two-hop neighboring node. When detecting the F1 fault, both the existence of significant differences between the interval numbers of the faulty node and its two-hop neighboring nodes (denoted as $S_{C^{S}}$ and $S_{L}$) and the cumulative sum of residuals between the interval numbers of the faulty node and its two-hop neighboring nodes (denoted as $S_{C^{S}}^{\prime }$ and $S_{L}^{\prime }$) are counted.
**Algorithm** **5:** Calculating the interval numbers of $C^{S}$ and *L* of node $S_{i}$.
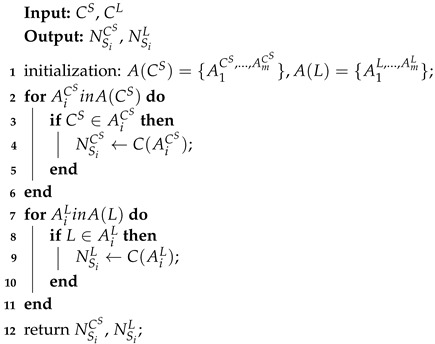


**Algorithm** **6:** Fault detection according to significant difference $S_{C^{S}}$, $S_{L}^{\prime }$, and cumulative sum of differences in interval numbering residuals $S_{C^{S}}^{\prime }$, $S_{L}^{\prime }$.

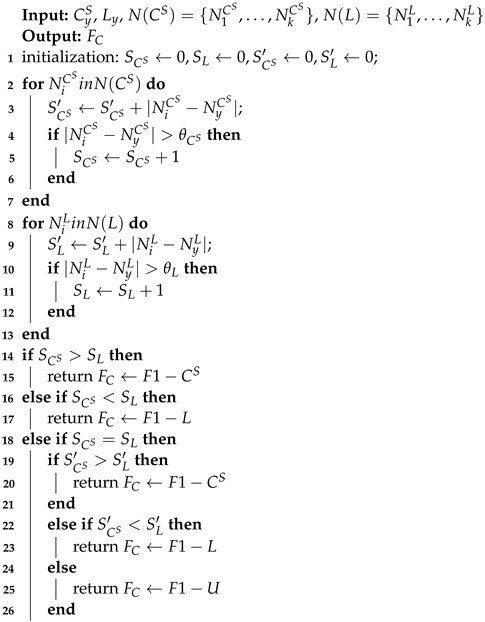



When the significant difference between the $C^{S}$ interval number of the faulty node and its two-hop neighboring nodes exceeds the difference between the *L* interval numbers, there is an abnormal solar panel current value and the fault label $F1-C^{S}$ is uploaded to the back end. When the significant difference between the $C^{S}$ interval number of the faulty node and its two-hop neighboring nodes is lower than the difference between the *L* interval numbers, there is an abnormal light intensity value and the fault label F1−L is uploaded to the back end. When $S_{C^{S}}=S_{L}$, the decision is made by the cumulative sum of residuals of interval numbers. When the accumulated sum of residuals of the faulty node and its two-hop neighboring nodes $C^{S}$ exceeds *L*, there is an abnormal solar panel current value. When the cumulative sum of the residuals of the faulty node and its two-hop neighboring nodes $C^{S}$ is less than that of *L*, there is an abnormal light intensity value. When the cumulative sum of the residuals of $C^{S}$ and *L* of the faulty node and its two-hop neighboring nodes are the same, the fault reason is uncertain and fault label F1-U is uploaded.

Based on these three steps, the proposed method performs fault detection without specifying a predefined threshold by transmitting only two-hop neighboring nodes $C^{S}$ and the *L* interval number (sent as an unsigned char data type in C and occupying only one byte) instead of the original data value. Therefore, the proposed method reduces the additional communication overhead and the dependence of the method on empirical threshold settings. In addition, the dual detection method [35] based on significant interval differences and cumulative sum of residuals can detect both cases of significant inconsistency in the trend of the faulty node and its two-hop neighboring nodes, as well as cases where the differences are small.

### 4.4. Feature Residuals Based Fault Detection

Although sending interval numbers through neighboring nodes rather than directly sending feature values can reduce the amount of data transmission, using interval number residuals as a basis for distributed fault detection could weaken the fault characteristics when there are not significant differences in fault-related features between nodes. Therefore, the analysis can be performed directly on the differences between the fault-related characteristics. For example, the mismatch between air temperature and temperature values inside the electrical box is difficult to quantize using interval numbers or operating status labels for the corresponding data. Considering the spatial correlation of temperature data in the case of geographical proximity, a decision can be made based on the residual difference between the temperature values of the faulty node and the neighboring nodes. As shown in Figure 9, the trends and residual values of air temperature and temperature inside the electrical box at different nodes are relatively small, thus, fault detection can be detected by the residual values between the faulty node and neighboring node features. Because of the low deployment density of SIL nodes, the two-hop neighboring nodes of a faulty node may only be one or two, and it is difficult to obtain good performance in distributed fault detection by voting. The residuals of air temperature values and temperature values inside the electrical box can be compared to further detect the F2 fault.

Because the fault is determined directly by the residual value of the temperature, the interval significant difference determination in the flow is not performed. The detection process is shown in Algorithm 7. The input data are the air temperature value $T_{y}^{0}$ of the faulty node, the temperature value in the electrical box $T_{y}^{1}$ and the air temperature $D\left(T^{0}\right)=T_{1}^{0},\ldots ,T_{k}^{0}$ and the temperature value in the electrical box $D\left(T^{1}\right)=T_{1}^{1},\ldots ,T_{k}^{1}$ of the two-hop neighboring nodes. When the cumulative sum of $T^{0}$’s residuals of the faulty node and its two-hop neighboring nodes exceeds the cumulative sum of $T^{1}$’s residuals, the result is that the air temperature sensor data are abnormal and the fault label $F2-T^{0}$ is uploaded to the back end. When the accumulated sum of $T^{0}$’s residuals of the faulty node and its two-hop neighboring node is less than the accumulated sum of $T^{1}$ residuals, the result is that the temperature value in the electrical box is abnormal and the fault label $F2-T^{1}$ is uploaded to the back end. When the accumulated sum of residuals of the faulty node and its two-hop neighboring nodes $T^{0}$ and $T^{1}$ are the same, the result is uncertain, and the corresponding fault label is F2−U.
**Algorithm** **7:** Fault detection according to cumulative sum of $T^{0}$’s and $T^{1}$’s residuals $S_{T^{0}}$, $S_{T^{1}}$.
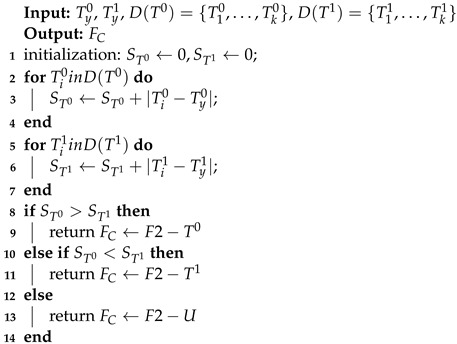


## 5. Experimental Setup

### 5.1. Experiment

The hardware implementation for the experimental setup is shown in Figure 10. Our network consists of 7 SIL-IoTs nodes, and we run the experiment from August to October 2021. The data sampling interval is 5 s. Except for the faults due to unexpected factors, we set up the following fault experiments on these nodes:Cover the light intensity sensor or solar panel with strong and weak shading plastic and sensor faults to simulate the mismatch between *L* and $C^{S}$.Disconnect the power supply or insert false data into the temperature sensor readings to simulate the mismatch between $T^{0}$ and $T^{1}$.Reboot the clock ship or install damaged modules to simulate the fault that the SIL is not switched on according to schedule.

The fault labels are shown in Table 3, where F1 indicates the root cause of the mismatch between the solar panel current value and the light intensity value, which can be divided into the abnormal solar panel current value (denoted as $F1-C^{S}$) and the abnormal light intensity value (denoted as F1−L), respectively. F2 indicates the mismatch between the air temperature value and the temperature value inside the electrical box, which can be divided into the abnormal air temperature value (denoted as $F2-T^{0}$) and the abnormal temperature value inside the electrical box (denoted as $F2-T^{1}$), respectively. F3 indicates the root cause of the SIL not turning on the lure lamp and the high-voltage metal mesh at the scheduled time, which can be divided into the abnormal light intensity value (denoted as F3−L) and an abnormal clock chip (F3−W), respectively.

Data from 0:00 on 1 September 2021 to 0:00 on 8 September 2021 are selected as historical data to obtain the parameters of the proposed method. There are 120,919 pieces of data per node, some of which are lost due to node maintenance. In addition, data from 0:00 on 9 September 2021 to 0:00 on 14 September 2021 are selected as test data, and each node has 86371 pieces of data.

The performance of different methods is estimated by ten-fold validation, where each validation selects 50% test data to verify the proposed method and ensure the reliability of the results. All methods are simulated on a PC with Windows 10 operating system, Intel Core i5-10400 CPU, and 16 GB RAM. In the simulation phase, all methods are written in Python 3.8. Then, the proposed methods are written in C and embedded in Arduino of SIL-IoTs node to estimate the energy consumption.

### 5.2. Comparison Method and Performance Indicators

In our experiments, four fault-detection methods designed for outdoor IoTs modules are compared, namely the DFD method [14], TinyD2 method [7], DSFD method [16], and TrusDet method [19]. All these methods are introduced and discussed earlier, where the DFD and DSFD methods adopt a voting strategy. The TinyD2 and TrusDet methods use a regression strategy. All methods use default parameters and are compiled using Python 3.8 and implemented on a PC with Windows 10 operating system, Intel Core i7-1165G7 2.8 GHz CPU, and 16 GM RAM. Assuming the fault status is positive, to evaluate the performance of the different methods, detecting accuracy (the proportion of correct results predicted by the model), false alarm rate (the probability of detecting fault-free data as faulty data), and missing alarm rate (the probability of detecting faulty data as fault-free data) is used. They are defined as:(9)$$Accuracy=\frac{TP+TN}{TP+TN+FP+FN}$$
(10)$$FAR=\frac{FP}{TN+FP}$$
(11)$$MAR=\frac{FN}{TP+FN}$$
where TP, TN, FP, and FN denote true positive, true negative, false positive (fault-free sample estimated as fault sample), and false negative (fault sample estimated as fault-free sample) samples, respectively.

## 6. Performance Evaluation

To demonstrate the effectiveness of the method, this section deals with it in three parts. First, this section analyses the spatial correlation of seven nodes to ensure the feasibility of distributed fault detection through neighboring device information. In addition, this section discusses the correlation of different target features to assess the degree of correlation between features. Second, this section evaluates the accuracy metrics of the proposed method and the comparison method for different numbers of neighboring nodes. Finally, this section shows the energy consumption of the proposed method through theoretical discussion and experiments.

### 6.1. Correlation Analysis

#### 6.1.1. Spatial Correlation Analysis

As shown in Figure 11, this section analyses the spatial correlation of 7 nodes based on historical data, where $\left[N_{1},N_{2},\ldots ,N_{7}\right]$ represents the device IDs of 7 SIL-IoTs nodes in Figure 10. In general, a Pearson correlation coefficient greater than 0.5 indicates a high spatial correlation between the two features, while a coefficient above 0.8 indicates a high spatial correlation. The results show that the features of all 7 SIL-IoTs nodes have a high positive spatial correlation. The spatial correlation of the high-voltage metal mesh current values is relatively low because the values are influenced by the random discharge of each node. The high-voltage metal mesh current values are only used to determine the operating status of SIL, and the data input to F3 fault detection is labeled as 0 or 1, so the relatively low spatial correlation has little impact. the reason for the low correlation between the N2 node solar panel currents and the other devices is that the N2 node deployment location is obscured by buildings and trees.

#### 6.1.2. Feature Correlation Analysis

Figure 12 shows the degree of model fit and the contribution of the most relevant features between the distributed fault-detection target features and other features in this section. The lure lamp current values and high-voltage metal mesh current values are only used to determine the operating state and not for residual analysis, thus they are not analyzed for inter-feature correlation.

As demonstrated from the goodness-of-fit in Figure 12, the goodness-of-fit $r^{2}$ for all indicators fluctuates between 98% and 100%, indicating that the target features can be accurately predicted by other highly correlated features. In addition, the blue line in Figure 12 shows the contribution of the features corresponding to the air temperature $(T^{0}$) and the temperature inside the electrical box ($T^{1}$) and the solar panel current value ($C^{S}$) and the light intensity value (*L*). The results show that $T^{0}$ and $T^{1}$ are the most correlated features and $C^{S}$ and *L* are the most correlated features, and that the contribution of each of these features to the corresponding feature exceeds 70%, i.e., the correlation between the features is strong and can be used for residual comparison.

#### 6.1.3. Variance Analysis

The variability of the historical data of the above 7 nodes is analyzed by the quartile method. The degree of variability of fault-related characteristics between the nodes is shown in Equation (Equation 12), where $x_{i}$ and $x_{j}$ denote the values at the same quartile for node *i* and node *j*, respectively.
(12)$$d_{i,j}=2\frac{x_{i}-x_{j}}{x_{i}+x_{j}}$$

Data with solar panel current values below 100 mA and light intensity values below 10,000 Lux are excluded to refine the distribution of data for 7 devices. As shown in Figure 13, the variance between air temperature and temperature inside the electrical box is significantly lower than 10%, while the variance between solar panel current and light intensity is significantly higher than 10%. The mean quartile variance between the different nodes of the air temperature is only 1.91%, indicating that the air temperature values at the different nodes have a relatively similar distribution trend. Similarly, the mean quartile variation between nodes for the temperature inside the electrical box is 3.46%, indicating a similar trend in the distribution of this feature between nodes. The mean quantile differences of 19.96% and 12.87% for solar panel current and light intensity indicate that there are high distribution differences between these two fault features, making it difficult to detect the F1 fault directly through the residuals between nodes. Since the lure lamp current values and high-voltage metal mesh current values are adapted to analyze differences in operating conditions, thus they are not analyzed for variance.

### 6.2. Influence of Quantile Parameters on the Mismatch between Solar Panel Current Values and Light Intensity Values

The choice of a different number of quantile numbers affects the accuracy of the $C^{S}$ and *L* interval mapping. When the number of quantiles is larger, $C^{S}$ and *L* can be finely divided into more intervals, which contributes to more accurately mapping $C^{S}$ and *L* to the corresponding interval number. However, the increase in the number of quantile numbers leads to more storage and computational resource consumption for the distributed fault-detection method. Therefore, this section sets the quantile parameters to [4, 6, 8, 10] for detecting $C^{S}$ and *L* mismatch faults. In addition, when the quantile parameter is increased, the detecting threshold for significant differences in interval numbers also affects the performance of distributed fault detection, thus this section sets the detecting threshold to no more than half of the quantile parameter. Figure 14, Figure 15, Figure 16 and Figure 17 show the quantile values of $C^{S}$ and *L* for different quantile parameters. Since $C^{S}$ and *L* are close to 0 for long periods of time at night, the data close to 0 are divided into separate intervals.

As illustrated from Figure 14, Figure 15, Figure 16 and Figure 17, the $C^{S}$ and *L* intervals for different nodes vary significantly, with the results as significant when deciles are used. For example, the values of the same decile of $C^{S}$ and *L* for nodes 2, 5, and 6 are significantly smaller than the other nodes, mainly because these three nodes are more affected by the environment. In addition, the light intensity sensor is strongly influenced by the translucency of the transparent housing. Nodes 6 and 7 consider the water’s edge, which is strongly influenced by humidity, and the transparent housing is susceptible to soiling, resulting in low light intensity sensor values. These factors are difficult to avoid when the SIL-IoTs node is actually deployed, thus this section does not screen out such situations to restore the situation when the SIL-IoTs node is actually deployed and disturbed by environmental factors.

Figure 18 and Figure 19 show the results of the proposed method for the F1 fault for different quantile intervals and different threshold conditions. The results show that the parameter setting “10-3” or “10-2” for the F1 fault can achieve a high detection accuracy. The horizontal coordinates indicate the number of different quantile intervals and different threshold conditions, e.g., “4-1” indicates that the fault detection is based on four segment intervals and a threshold condition where the residual value between the fault node and the two-hop neighboring nodes intervals is greater than 1. According to Figure 18, the best results for $F1-C^{S}$ fault are obtained for ten segment intervals and interval numbering residuals ($\theta (C^{S})$ in Algorithm 5) greater than 3, while the best results for F1−L fault are obtained for ten segment intervals and interval numbering residuals ($\theta_{L}$ in Algorithm 5) greater than 2. As the number of segment intervals increases, the detection accuracy and F1-score of the F1 fault increases; however, the increase in the interval numbering residual detecting threshold is not necessarily beneficial to the detecting accuracy and F1-score.

Figure 19 demonstrates the performance of the false alarm rate and the missing alarm rate for different quantile intervals as well as for different threshold conditions. Consistent with the detecting accuracy and F1-score results in Figure 18, the best trade-offs are achieved for $F1-C^{S}$ fault when ten inter-quartile intervals are used and inter-quartile numbering residuals are greater than 3, and for F1 − L fault when ten inter-quartile intervals are used and inter-quartile numbering residuals are greater than 2. The false alarm rate and the missing alarm rate change as the need for fault-detection sensitivity changes. Therefore, the appropriate fault-detection parameters can be selected based on a trade-off between the two to meet the needs of different scenarios. In the case of distributed fault detection for SIL-IoTs, either “10-2” or “10-3” can be used for the F1 fault to achieve good performance. In this section, the “10-3” parameter setting is used subsequently.

### 6.3. Accuracy of Different Methods

This section compares the proposed distributed fault-detection methods and the accuracy metrics of the four compared methods. The metrics of the proposed method are shown as blue bar graphs in Figure 20. Figure 20a shows that the proposed method has the highest accuracy for all fault categories except F2, while the other methods fail to detect $F2-T^{0}$ fault effectively, indicating that setting detecting thresholds and comparing them to residual values is not conducive to detecting abnormal temperature values inside the electrical box.

The above results show that the proposed methods can achieve good performance for faults F1, F2, and F3. In addition, the fault SIL-IoTs nodes only have one to three two-hop neighboring nodes, thus it is difficult to obtain good performance using the voting method for distributed fault detection. Detecting faults by way of interval numbering residuals or feature value residuals not only reduces the dependence on the setting of threshold parameters but also avoids the difficulty in detecting faults when there are not enough neighboring nodes.

### 6.4. Impact of Different Numbers of Neighboring Nodes

To investigate the impact of different numbers of neighboring nodes on the distributed fault-detection method, this section compares the performance of the proposed method and comparison methods based on one-hop neighboring nodes with those based on two-hop neighboring nodes. The results show that the proposed method achieves the highest detecting accuracy and F1-score under both the one-hop and two-hop neighboring node conditions.

As shown in Figure 21, the metrics of the different methods based on one-hop neighboring nodes are represented as bar charts with narrower dashed borders, where the number of neighboring nodes used for fault detecting based on one-hop neighboring nodes is fewer than or equal to the case when it is based on two-hop neighboring nodes. The proposed method and the DFD method show a slight decrease in detecting accuracy and F1-score when the number of neighboring nodes is reduced, which indicates that the proposed method has some dependency on the number of neighboring nodes. The results of the TinyD2 method show some improvement in detecting accuracy and F1-score when the number of neighboring nodes is reduced, which results from the exclusion of information about neighboring nodes with similar states. In summary, the proposed method achieves the best results compared to comparison methods in both the one-hop and two-hop neighboring node cases.

### 6.5. Lightweight Analysis of the Proposed Method

Due to the limited computational resources of the SIL-IoTs (using Arduino with 20 MHz CPU speed, 32 KB of program memory, and 2 KB RAM size for node-level decision-making), the proposed method should be lightweight. Based on this, the proposed method is computationally simple, requiring only a small number of parameters to be pre-stored in each node, as well as a simple accumulation and calculation to obtain the detection results, helping to reduce the ratio of computational to storage capacity of the control chip.

To evaluate the lightweight performance of the proposed method, this section deploys the proposed method on an Arduino chip using a C program. The original program, which does not participate in compile-time data acquisition etc., occupies 19.6% of RAM (402 bytes) and 17.5% of Flash (5656 bytes) in the Arduino. When the proposed method is added, the program takes up 20.6% of RAM (420 bytes) and 20.6% of Flash (6644 bytes) in the Arduino. Therefore, the proposed method uses an additional 0.9% of RAM (18 bytes) and 3.1% of Flash (988 bytes), which has little impact on the Arduino.

### 6.6. Energy Consumption of the Proposed Method

Due to the limited energy of the SIL-IoTs node, this section evaluates the proposed method in terms of data transfer energy consumption as well as the energy consumption of the proposed method running on the Arduino. The argumentation and experimental results show that the proposed method has low energy consumption.

The proposed method detects F1 and F3 faults by requiring the neighboring nodes to send either the interval number or the operating condition information (represented in C by unsigned char, i.e., one byte) to the faulty node, thus reducing the additional communication overhead and energy consumption caused by the transmission of sensor measurements (floating-point data). Assuming that the faulty node has an F1 fault and there are *k* two-hop neighboring nodes, the energy required to transmit the two floating-point data to the faulty node by other distributed methods can be calculated using the data transmission energy formula [36]:(13)$$E_{t-ij}=n\times \left(\alpha_{12}+\alpha_{2}d_{ij}^{k}\right)$$
where $\alpha_{12}$ and $\alpha_{2}$ are the energy consumption parameters of the base band and amplification circuits of the sending node, $d_{ij}$ denotes the distance from node *i* to node *j*, *k* is the propagation path attenuation factor, typically an integer between 2 and 4, and *n* denotes the length of the data to be sent. A total of 4 bytes of floating-point-type data need to be sent for each two-hop neighboring node when detecting F1 and F3 faults. Therefore, the total energy consumption of the other methods to perform one data transmission is $4\times k\times (\alpha_{12}+\alpha_{2}d_{ij}^{k})$. The faulty node also consumes energy to receive the data, which is calculated as:(14)$$E_{r}=n\times \alpha_{11}$$
where $\alpha_{11}$ denotes the receiving node circuit energy consumption parameter. Therefore, the total energy consumption of the other methods to perform one data transmission is $4\times k\times \alpha_{11}$. Since the proposed method only requires two bytes of data per two-hop neighboring node when detecting F1 and F3 faults, the total energy consumption in terms of data transmission is one quarter of that of the other methods. Similarly, the energy consumption of the proposed method is also one quarter of that of the other methods in terms of data transmission.

To accurately calculate the energy consumption of the proposed method in the Arduino chip, this section uses an AC/DC-type electrical parameter meter to measure the total energy consumption of performing 10,000 distributed fault detection. The experimental apparatus and circuit connections used to measure the energy consumption of the proposed method are shown in Figure 22, where the brand of the AC/DC-type electrical parameter measuring instrument used is model PM9200, Napui Electronic Technology Co., Ltd., Dongguan, China. To ensure the stability of the power supply, a triple-channel DC benchtop power supply, brand Keithley (Tektronix, Berkshire, UK), model 2231A-30-3, is used in the experiment. the red line in the figure is the positive pole, the blue line is the negative pole, and the black dotted line indicates the direction of data transmission. The energy consumption data monitored by the electrical parameter meter is transferred to the computer via USB and stored in the relevant software developed by the manufacturer.

To reduce the impact of sensor data acquisition and other functions on the energy evaluation, no sensors are added to the PCB and only data written in advance is used for detection in the experiments. To improve the reliability of the results, we repeat three times with and without the proposed method. The experimental results are based on the scenario where the faulty node has three two-hop neighboring nodes, which is a case with many neighboring nodes in this scenario. The results of the multiple experiments are shown in Table 4. The average total active power when running the proposed method is 1.1724 mWh, while the average total active power when not running the proposed method is 1.1672 mWh. The additional active power consumed by running the proposed method 10,000 times is 0.0053 mWh, which is 0.45% of the total active power when not running the proposed method. In terms of battery capacity, the proposed method consumes an additional $4.67\times 10^{-4}$ AH to run 10,000 times. The 12 V, 38 AH battery of SIL-IoTs nodes used in this paper can be used to run the proposed method over 813 million times. In summary, the proposed method proposed is suitable for SIL-IoTs because of its low energy consumption for operation.

## 7. Conclusions

In this study, a fault-detection scheme for SIL-IoTs is proposed to address faults that cannot be estimated by single-node information. Based on the experimental results, the following conclusions are drawn: (1) The proposed method achieves an average F1-score of 92.42% and 95.59% based on one-hop and two-hop neighboring nodes, respectively, demonstrating high performance in fault detection. (2) When compared to existing methods, the proposed method outperforms them significantly, with an average F1-score improvement of at least 48.65%. This highlights the superiority of using the cumulative sum of residuals over traditional approaches involving threshold setting or single-feature comparison. (3) The demonstration and experiments reveal that the proposed method reduces the energy consumption of data transmission for information interaction between nodes by 25%. Moreover, the additional energy consumption on the Arduino chip is minimal, accounting for only 0.27% of the total power.

The above advantages demonstrate that the proposed method performs well in detecting different fault types and accommodating varying numbers of neighboring nodes. Additionally, the method is lightweight in terms of energy consumption, parameter usage, and system resources when implemented on the Arduino chip. Thus, it fulfills the need for an efficient and resource-friendly distributed fault-detection method.

It is worth noting that the proposed method may not achieve 100% detection accuracy due to the presence of certain noise signals, such as sensor faults and electromagnetic interference caused by high-voltage discharge [5]. Detecting these types of noise signals can be challenging. To overcome this limitation, future work could explore the fault-detection scheme’s performance under conditions of low reliability in data acquisition and transmission. This would involve investigating methods to improve fault detection in the presence of such challenging noise signals. Furthermore, the proposed method relies heavily on historical data and prior knowledge. Future research efforts could focus on designing a highly adaptive algorithm capable of self-learning fault characteristics even with limited historical data. This would enable the system to continuously improve its fault-detection capabilities and adapt to changing environmental conditions. By addressing these potential future directions, further advancements can be made to enhance the robustness and adaptability of the fault-detection scheme for SIL-IoTs.

## Figures and Tables

**Figure 1 sensors-23-06672-f001:**
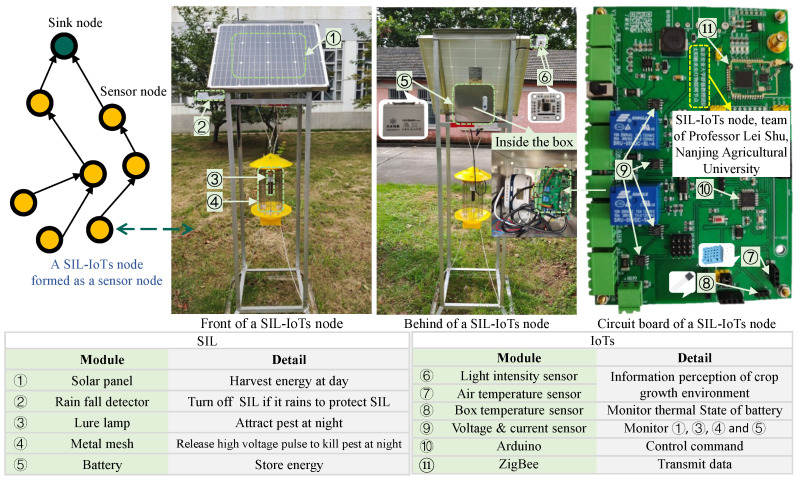
An example of a SIL-IoTs node, where a temperature sensor inside an electrical box is used to monitor the thermal state of the battery and IoTs devices. The light intensity sensor is used to monitor the condition of solar panels. More details can be seen from [4].

**Figure 2 sensors-23-06672-f002:**
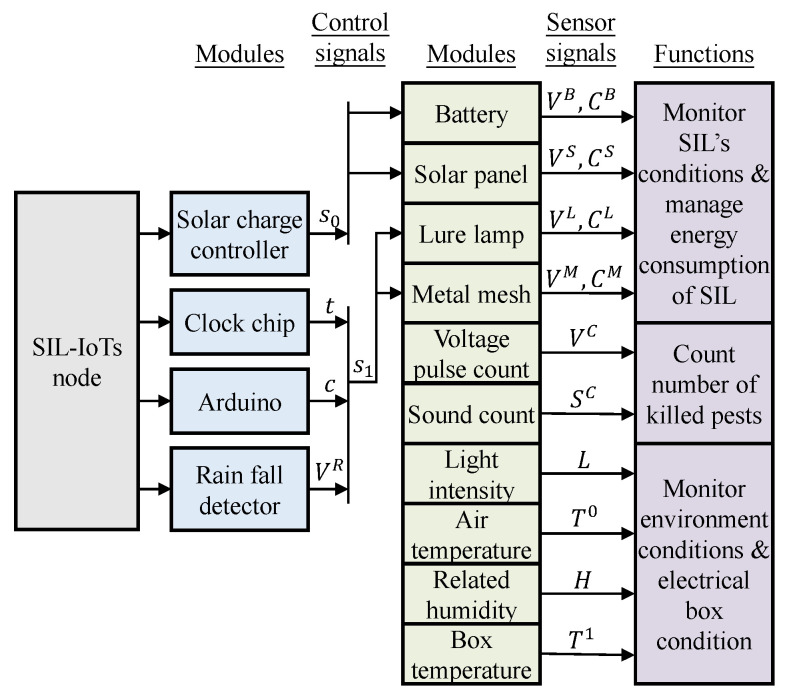
The SIL-IoTs system, where parts of the sensor signals are analyzed in Arduino and transmitted to the cloud server via ZigBee.

**Figure 3 sensors-23-06672-f003:**
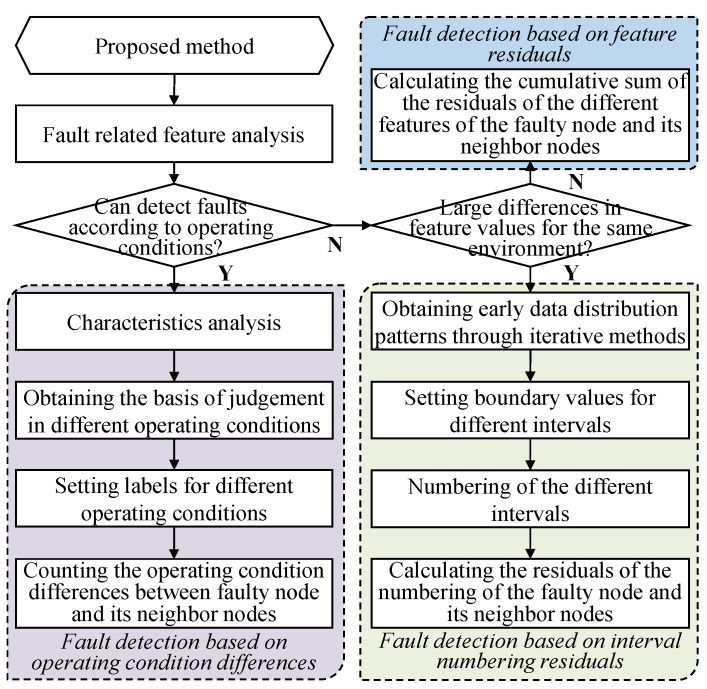
Flowchart of the proposed method.

**Figure 4 sensors-23-06672-f004:**
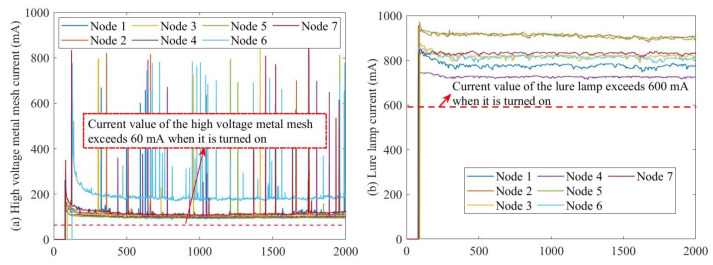
Current values of (**a**) high-voltage metal mesh and (**b**) lure lamp.

**Figure 5 sensors-23-06672-f005:**
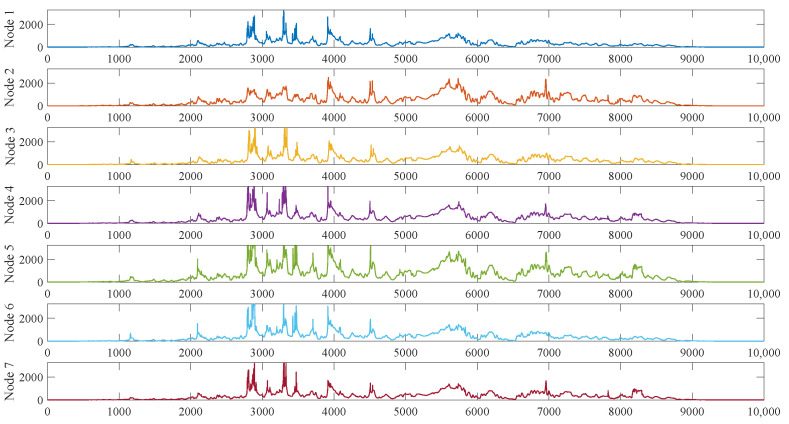
Solar panels current values of different nodes with similar tendencies.

**Figure 6 sensors-23-06672-f006:**
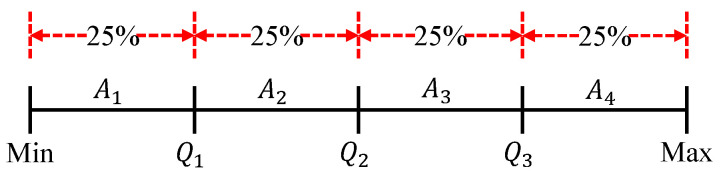
Diagram of quartile method.

**Figure 7 sensors-23-06672-f007:**
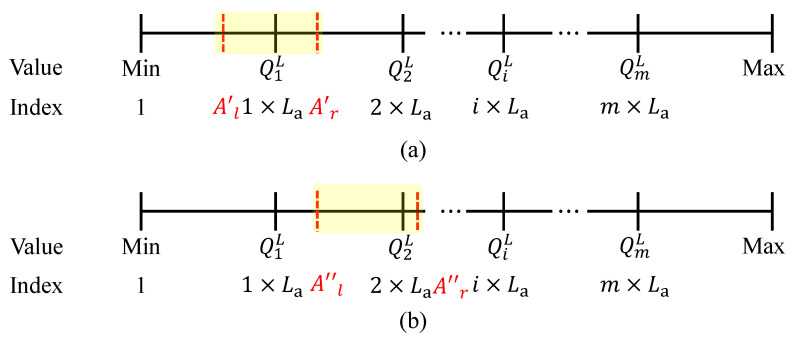
Accumulate interval statistics until $A_{r}$ is greater than $2\times L_{a}$, where black lines represent quantile index values and red dashed lines represent interval statistics. (**a**) $A_{r}<2\times L_{a}$, the current cumulative index value does not exceed the quantile index value, and the next index value needs to be accumulated. (**b**) $A_{r}\geq 2\times L_{a}$, The current cumulative index value does not exceed the quantile index value, and the next index value needs to be accumulated.

**Figure 8 sensors-23-06672-f008:**
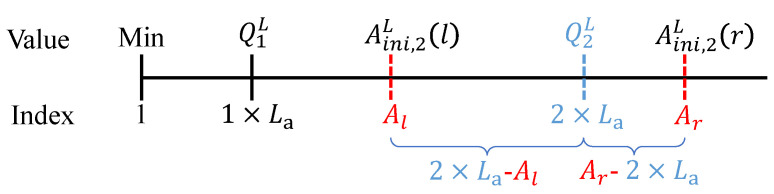
Calculate $Q_{2}^{L}$ according to $A{\left(ini,2\right)}^{L}\left(r\right)$, where red dashed lines represent interval statistics used to calculate $Q_{2}^{L}$ and blue line represents quantile index value.

**Figure 9 sensors-23-06672-f009:**
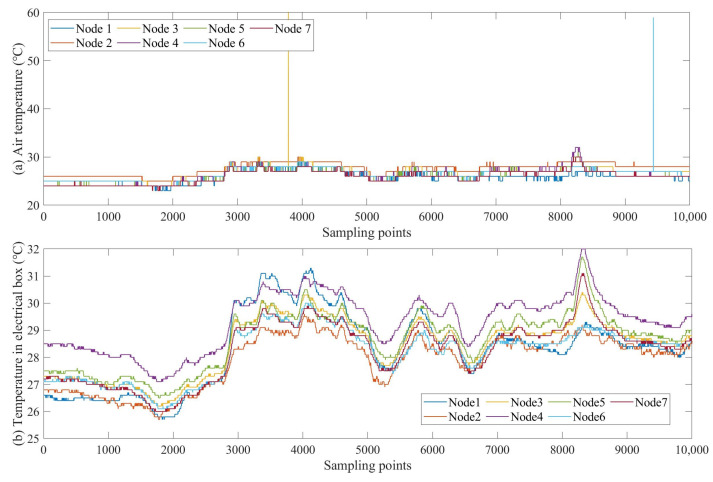
(**a**) Variation trends of air temperature at different nodes are similar, and (**b**) variation trends of temperature values inside the electrical box are similar.

**Figure 10 sensors-23-06672-f010:**
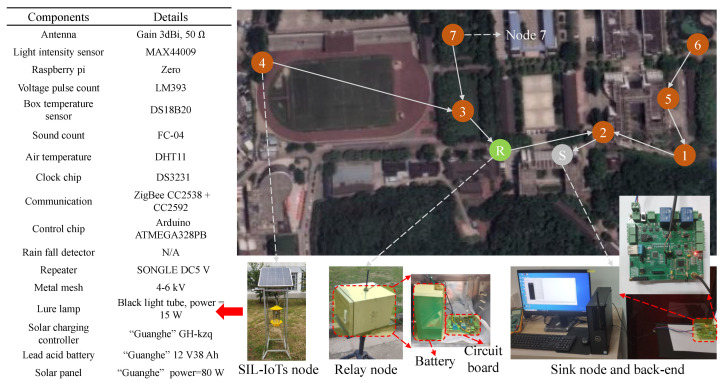
A real-world application of 7 SIL-IoTs nodes, with white arrows indicating the direction of data propagation, red arrow indicating the components and details of a SIL-IoTs node, and red dashed arrows indicating the interior or details of corresponding components. White dashed arrows indicating the real-world devices corresponding to different dots, where the number in dots indicating device ID, R in the green dot indicating the relay node, and the S in grey dot indicating the sink node.

**Figure 11 sensors-23-06672-f011:**
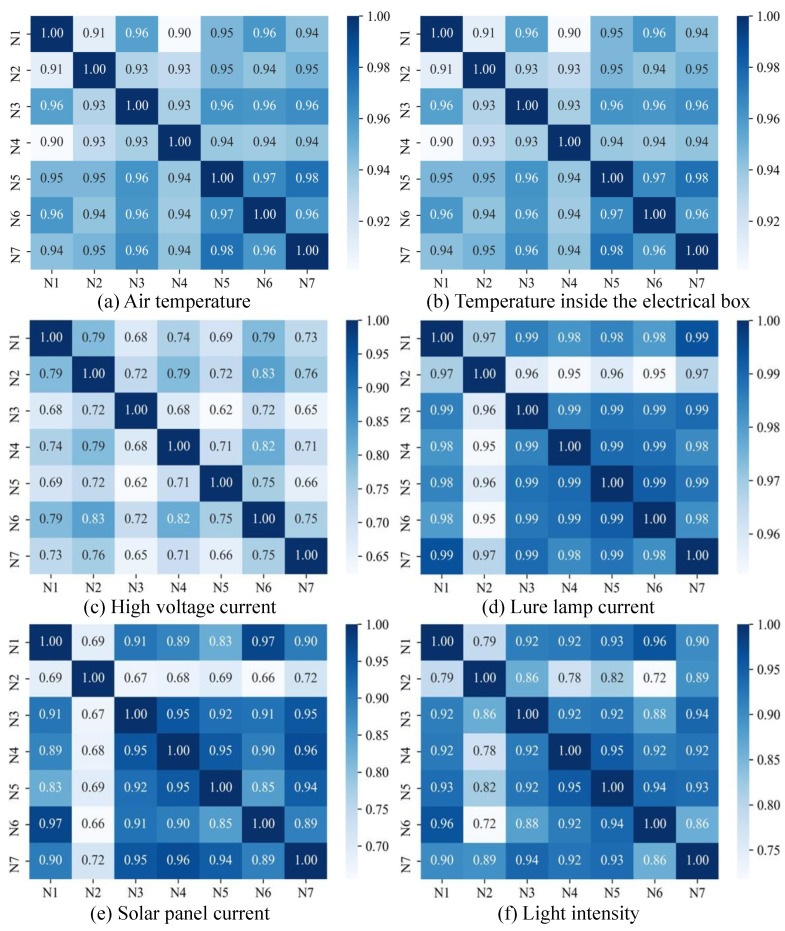
Spatial correlation analysis of features related to distributed fault detection.

**Figure 12 sensors-23-06672-f012:**
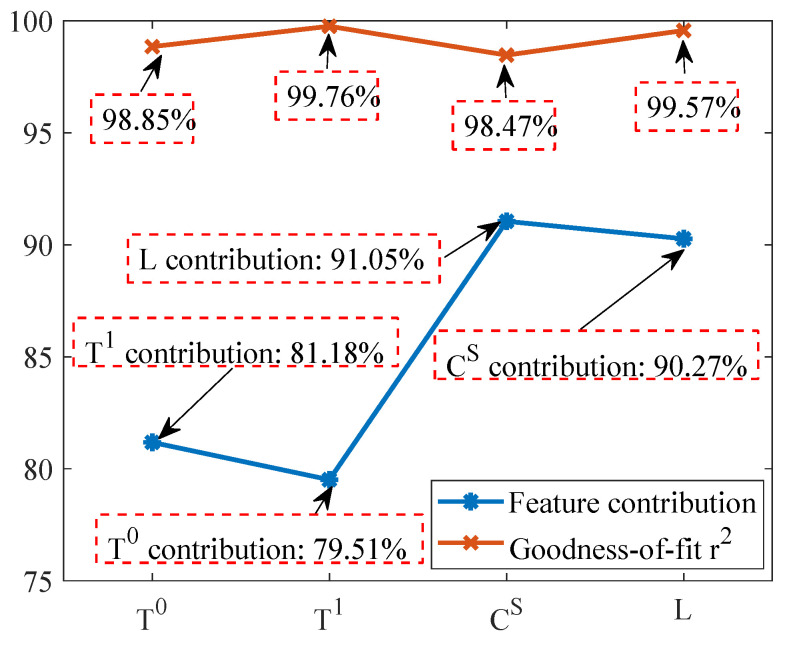
Correlation analysis results of four features in residual analysis.

**Figure 13 sensors-23-06672-f013:**
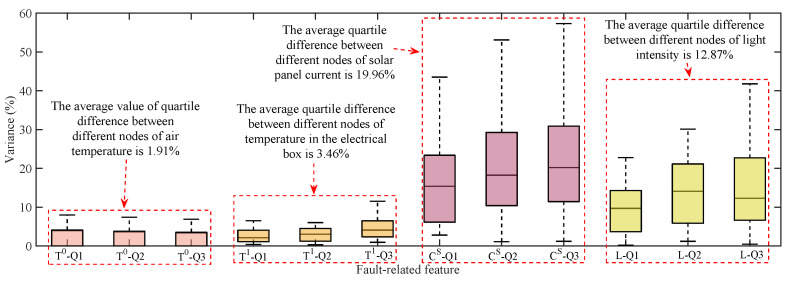
Historical data difference of 7 nodes expressed by quartile.

**Figure 14 sensors-23-06672-f014:**
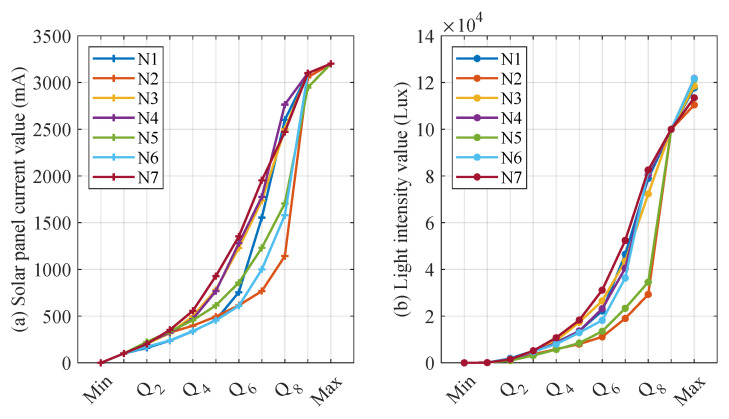
Parameter settings when iterating into ten intervals based on early data.

**Figure 15 sensors-23-06672-f015:**
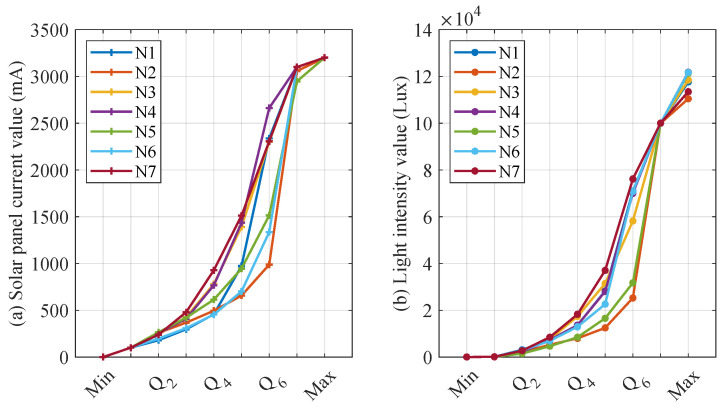
Parameter settings when iterating into eight intervals based on early data.

**Figure 16 sensors-23-06672-f016:**
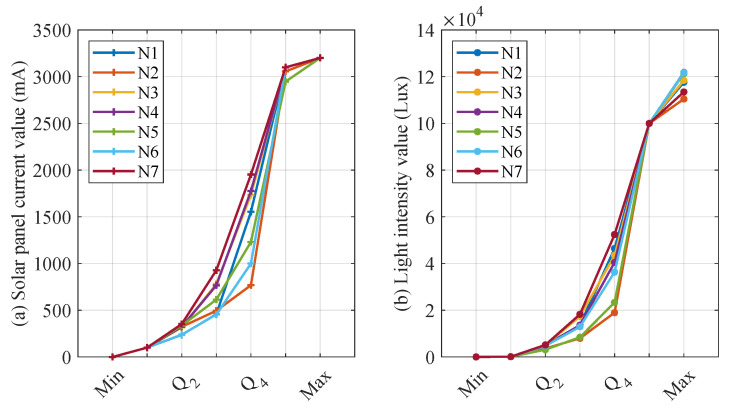
Parameter settings when iterating into six intervals based on early data.

**Figure 17 sensors-23-06672-f017:**
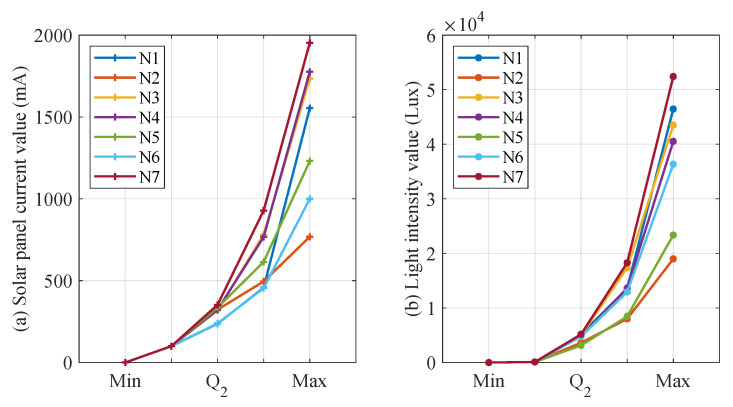
Parameter settings when iterating into four intervals based on early data.

**Figure 18 sensors-23-06672-f018:**
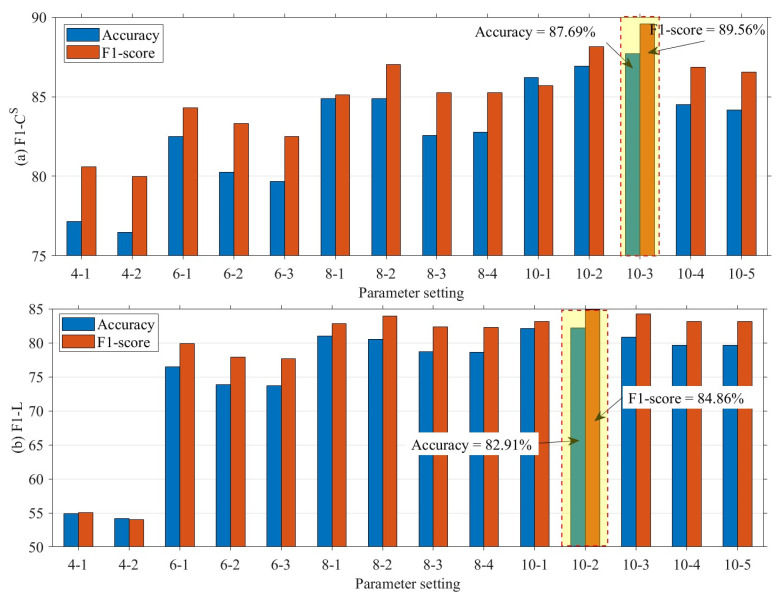
Detecting accuracy and F1-score of F1 fault under different parameters, where “4-1” denotes detecting F1 fault by 4 inter-quartile intervals and inter-quartile numbering residuals greater than 1.

**Figure 19 sensors-23-06672-f019:**
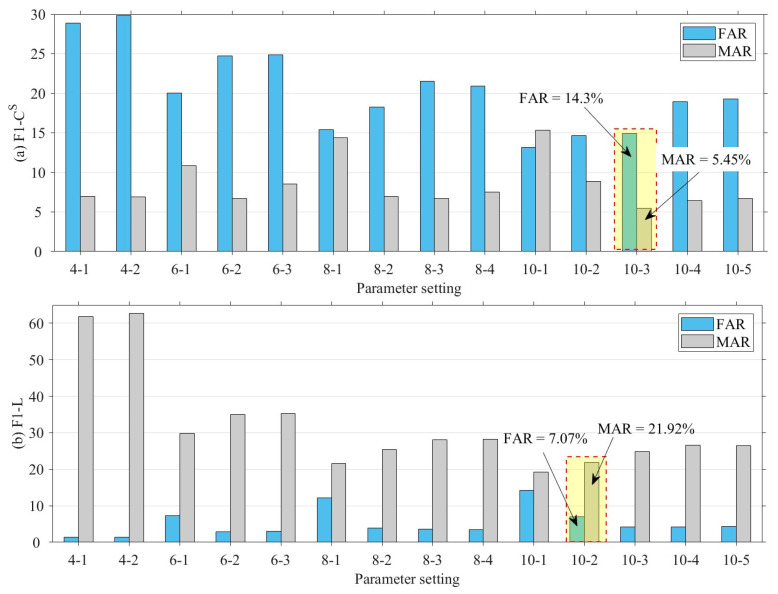
False alarm rate and missing alarm rate of F1 fault under different parameters.

**Figure 20 sensors-23-06672-f020:**
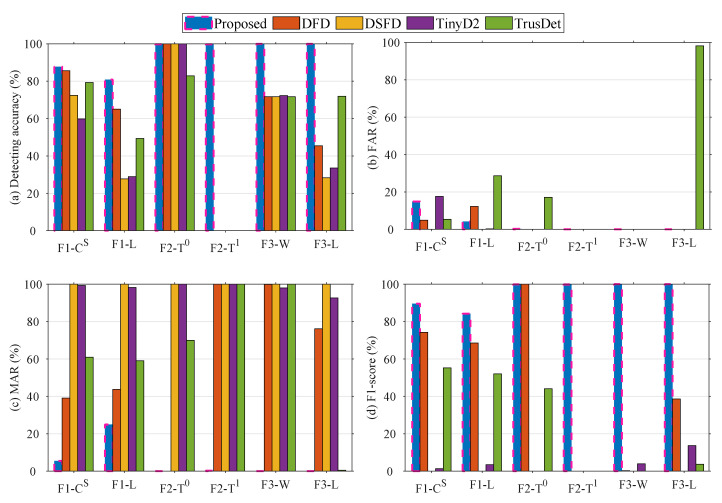
Distributed fault-detection performance of different methods.

**Figure 21 sensors-23-06672-f021:**
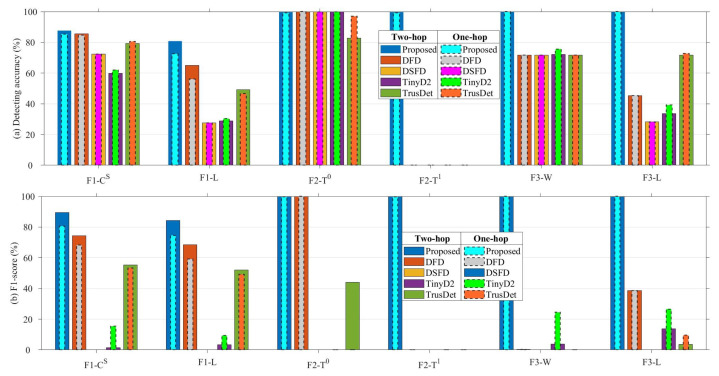
Performance of different methods based on one-hop and two-hop neighboring nodes.

**Figure 22 sensors-23-06672-f022:**
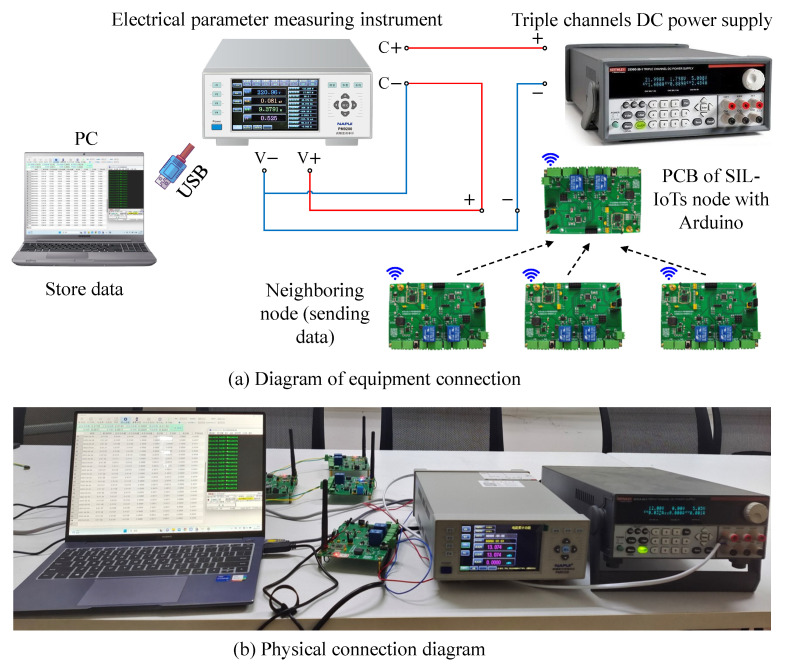
Schematic diagram and physical connection diagram of energy consumption experimental equipment for the proposed method.

**Table 1 sensors-23-06672-t001:** Comparison of SIL and SIL-IoTs node.

	SIL	SIL-IoTs
Price	CNY 1100 (about $160)	CNY 1500 (about $219)
Function	Harvest energy Kill pest	SIL’s functions Count killed pests Monitor component status Monitor environment
Advantage	Cheap Easy to use	Provide farmers with killed pest statistics for targeted pesticide usage Detect faults timely to ensure reliability of SIL-IoTs
Drawback	Inability to perceive information	Expensive price

**Table 2 sensors-23-06672-t002:** Comparison of research related to distributed fault detection.

Ref.	Scenario	Implement	Method	Deployment Density	Battery-Powered	Lightweight Design	Energy Consumption
[13]	Printer systems	Sensor node	Consistency check	N/A	N/A	N/A	N/A
[14]	WSNs	Simulation	Dual thresholds detection	1024/32 × 32 units	N/A	N/A	N/A
[15]	WSNs	Simulation	Improved dual thresholds detection	200/30 × 30 units	N/A	N/A	N/A
[7]	Canopy closure monitoring sensors	MSP430	Cumulative sum sliding window	200/2 $\times \;10^{6}$ m${}^{2}$	✓	N/A	N/A
[16]	WSNs	Simulation	Improved 3-σ test	1024/1 $\times \;10^{6}$ m${}^{2}$	✓	N/A	N/A
[17]	Industrial control systems	Simulation	Genetic algorithms	N/A	N/A	N/A	N/A
[18]	WSNs	Simulation	Support vector machines	200/30 × 30 units	N/A	N/A	N/A
[19]	WSNs	Simulation	Dual thresholds detection	1024/2.62 $\times \;10^{5}$ m${}^{2}$	N/A	N/A	N/A
[20]	Infrared sensors	Arduino	Exponential smoothing	N/A	✓	✓	N/A
[21]	WSNs	Simulation	Exponential smoothing and median value detection	N/A	✓	✓	N/A
	Our	Arduino	Quantile method and residual test	7/2.72 $\times \;10^{5}$ m${}^{2}$	✓	✓	✓

**Table 3 sensors-23-06672-t003:** Labeling of faults.

Fault Type	Label	Measurement
Solar panel current abnormal	$F_{1}^{C^{S}}$	$C^{S}$
Light intensity sensor fault	$F_{1}^{L}$	*L*
Air temperature sensor fault	$F_{2}^{T^{0}}$	$T^{0}$
Box temperature sensor fault	$F_{2}^{T^{1}}$	$T^{1}$
Lamp current abnormal	$F_{3}^{C^{L}}$	$C^{L}$
Metal mesh current abnormal	$F_{3}^{C^{M}}$	$C^{M}$

**Table 4 sensors-23-06672-t004:** Energy consumption statistics of the proposed method in Arduino.

	Experimental Times	Total Active Energy (mWh)	Total Ah	Average Power (mW)
With the proposed method	1	1.1715	0.1021	0.4217
	2	1.1697	0.1019	0.4209
	3	1.1764	0.1025	0.4235
Without the proposed method	1	1.1663	0.1016	0.4197
	2	1.1679	0.1018	0.4202
	3	1.1675	0.1017	0.4201

## Data Availability

Not applicable.

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
