# Peer review of "A Lightweight Fault-Detection Scheme for Resource-Constrained Solar Insecticidal Lamp IoTs"

_sensors, 2023, doi:10.3390/s23156672_

Round 1
Reviewer 1 Report
Introduction:
The paper presents an innovative approach to ensure the dependability and safety of SIL-IoTs in agricultural settings. The study addresses the challenges posed by complex environmental changes and device deterioration, offering a sensor-level fault detection scheme that considers realistic constraints such as computational resources and energy. This positive report highlights the key contributions and strengths of the proposed method.
Summary of the Research:
The research paper introduces a distributed fault detection method that utilizes fault characteristics, operation condition differences, interval number residuals, and feature residuals. Through a series of well-designed experiments, the effectiveness of the proposed method was thoroughly validated. The results of these experiments demonstrated an impressive average F1-score of 95.59%.
Key Findings and Contributions:
One of the major findings of this study is the development of a lightweight fault detection scheme for SIL-IoTs that effectively addresses the challenges faced in agricultural settings. By considering constraints related to computational resources and energy consumption, the proposed method provides a practical and efficient solution.
The study's results also highlight the resource efficiency of the proposed method. With an additional power consumption of only 0.27% of the total power and utilization of 0.9% RAM and 3.1% Flash on the Arduino of the SIL-IoTs node, the method proves to be highly lightweight and energy-efficient. These findings are crucial as they demonstrate the feasibility of implementing the proposed fault detection scheme in real-world agricultural scenarios without compromising the overall performance and power requirements of the SIL-IoTs system.
Moreover, the paper provides a comprehensive analysis of fault characteristics, which allows for a more accurate and reliable detection of system failures. By incorporating operation condition differences, interval number residuals, and feature residuals into the fault detection scheme, the proposed method demonstrates its ability to effectively detect faults and ensure the dependability of SIL-IoTs.
Significance and Implications:
The proposed fault detection scheme presented in this paper has significant implications for the field of agricultural IoT. By addressing the challenges associated with complex environmental changes and device deterioration, the method contributes to the overall reliability and safety of SIL-IoTs. The high average F1-score achieved by the proposed method indicates its superior performance in fault detection.
Furthermore, the lightweight nature of the proposed scheme ensures that it can be implemented on resource-constrained devices, such as the Arduino of the SIL-IoTs node, without significant additional power consumption or memory utilization. This aspect is crucial in practical deployments where energy efficiency and resource optimization are paramount.
Conclusion:
In conclusion, the paper presents a valuable contribution to the field of agricultural IoT. The proposed method showcases remarkable fault detection capabilities, achieving an average F1-score of 95.59%. Additionally, the method's lightweight nature and minimal resource requirements make it highly practical and energy-efficient. This research opens new avenues for enhancing the dependability and safety of SIL-IoTs in pest monitoring, prediction, and prevention, thereby advancing the field of agricultural-enabled electronic devices. So, I recommend this paper for the publication.
Author Response
Thank you for reviewing our paper.
Reviewer 2 Report
In this study, the authors proposed A Lightweight Fault-Detection Scheme for Resource Constrained Solar Insecticidal Lamps Internet of Things (SIL-IoTs). For this, a sensor-level lightweight fault detection scheme is proposed that takes into account realistic constraints for example; computational resources and energy. The manuscript is well written. Some minor comments are given below.
* Please add motivation and motivation and the benefits of this research along with contribution in the introduction section.
*In related work please add the pros and the cons of recent studies.
* In the proposed work the explanation of equations 3 and 4 should be mentioned.
* algorithm 1 and Algorithm 2 explanation details are required.
* The details of the simulation environment and tool should be required in the results section.
*
The English are fine. However, typos, grammar, and mistakes should be checked and corrected in the final version.
Author Response
- Comment 1: Please add motivation and motivation and the benefits of this research along with contribution in the introduction section.
Answer 1: Thank you for reviewing our paper. Done. We have added motivation and benefits in Introduction section. The details:
- Page 3 Line 59-78: The motivation and benefits of this research are as follows: As SIL-IoTs nodes are often deployed in the field, they are susceptible to ageing, vandalism and other factors that can lead to failures. In order to detect faults in SIL-IoTs, appropriate fault diagnosis methods need to be investigated. Deploying fault diagnosis methods on the device side can improve the efficiency of device data usage and reduce the energy consumption of missing data and transmitted data due to data backhaul. The background characteristics of SIL-IoTs need to be considered when designing fault diagnosis methods, including:
- The computational burden of fault detection strategies needs careful considerations in practical applications. For example, SIL-IoTs nodes are resource-constrained devices, which indicates that the fault detection model should be lightweight to reduce the computational burden.
- The low deployment density of SIL-IoTs node leads to an insufficient number of nodes in geographical proximity, and the existing distributed fault diagnosis methods are difficult to achieve better results in this case, hence it is critical to design a distributed fault diagnosis method with low dependence on the number of neighboring nodes.
- SIL-IoTs is a kind of typical agricultural IoTs equipment, thus the proposed method in this paper can also be used in IoT equipment with similar characteristics in, e.g., intelligent irrigation equipment, micro weather stations.
- Comment 2: In related work please add the pros and the cons of recent studies.
Answer 2: Thank you for reviewing our paper. Done. We have added the pros and the cons of recent studies, the details:
- Page 5 Line 169-184: In summary, the advantages of recent studies include 1) avoiding large amounts of data transmission to the backend by means of local information decision making, and 2) avoiding inaccurate fault diagnosis results due to missing or asynchronous data from neighboring nodes when fault diagnosis is performed in the backend. It should be noted that, recent studies in Table 2 are based on scenarios with a high deployment density of sensor nodes, whereas the deployment density of SIL-IoTs nodes is usually sparse [2], which denotes that diagnosing fault by voting strategy can lead to a decrease in diagnostic accuracy. The literature [2] shows that when the effective pest killing range of SIL-IoTs nodes is 110 m (i.e., deploying SIL-IoTs nodes at 110 m intervals), only 10 nodes need to be deployed on a 600 m × 600 m map according to the optimal deployment method proposed in the literature. Compared to the literature [18], which deploys 1024 nodes on a 512 m × 512 m map, or the literature [16], which deploys 1024 nodes on a 1000 m × 1000 m map, the deployment density of SIL-IoTs nodes is significantly lower. In addition, distributed fault diagnosis methods require data interaction between nodes, which generates additional communication energy consumption, which is detrimental for SIL-IoTs nodes.
- Comment 3: In the proposed work the explanation of equations 3 and 4 should be mentioned.
Answer 3: Thank you for reviewing our paper. Done. We have added some content to make it clearer. The details:
- Page 9 Line 273-276, for equation 3: The spatial correlation between the device to be detected and its neighboring nodes can be obtained from equation 3, which represents the covariance of and divided by the product of the standard deviations of and .
- Page 9 Line 289-294, for equation 4: PI is obtained by randomly shuffling each feature and computing the change in the performance of random forest. As shown in equation 4, the importance score ranking is estimated by differences between the regression accuracy without randomly exchanging permuted out of bag data (denoted as ) and the regression accuracy with randomly exchanging permuted out of bag data (denoted as ).
- Comment 4: Algorithm 1 and Algorithm 2 explanation details are required.
Answer 4: Thank you for reviewing our paper. Done. We have added some content to make it clearer. The details:
- Page 10 Line 321-324, for Algorithm 1: The lure lamp and high voltage metal mesh will only be switched on if the node determines that it is currently night time based on the local time data. At this time, should greater than 60 mA and should greater than 600 mA. Therefore, will be set to 1 and 0 otherwise.
- Page 10 Line 327-331, for Algorithm 1: As SIL-IoTs nodes are deployed in agricultural fields, they should not be able to receive illumination from external light sources such as street lamps at night, and their light intensity value should be at a lower level. Therefore, when the light intensity value is below 10 Lux, it is judged to be currently in a night state.
- Page 10-11 Line 334-342, for Algorithm 2: and represent and of the node (faulty node). S( )={ ,..., }, S( )={ ,..., } denote the and of all two-hop neighboring nodes of the node , where indicates the number of two-hop neighboring nodes of the node . S( ) and S( ) denote the accumulated sum of and residuals of the faulty node and its two-hop neighboring nodes, where the initial values of both S( ) and S( ) are set to 0. The final value of S( ) is obtained by accumulating the absolute value of the residual value between and (each two-hop neighboring node of node ). The final value of S( ) is obtained by the same way.
- Comment 5: The details of the simulation environment and tool should be required in the results section
Answer 5: Thank you for reviewing our paper. Done. We have added content to make it clear. The details:
- Page 19 Line 543-546: All methods are simulated on a PC with Windows 10 operating system, Intel Core i5-10400 CPU and 16GB RAM. In simulation phase, all methods are written in Python 3.8. Then, the proposed methods are written in C and embedded in Arduino of SIL-IoTs node to estimate the energy consumption.
- Comment 6: The English are fine. However, typos, grammar, and mistakes should be checked and corrected in the final version.
Answer 6: Thank you for reviewing our paper. Done. We have carefully checked the typos, grammar and mistakes.
Please see the revised manuscript in the attachment.
